# Prognostication of chronic disorders of consciousness using brain functional networks and clinical characteristics

**Ming Song[1,2†], Yi Yang[3†], Jianghong He[3], Zhengyi Yang[1,2], Shan Yu[1,2], Qiuyou Xie[4], Xiaoyu Xia[3], Yuanyuan Dang[3], Qiang Zhang[3], Xinhuai Wu[5], Yue Cui[1,2], Bing Hou[1,2], Ronghao Yu[4], Ruxiang Xu[3]\*, Tianzi Jiang[1,2,6,7,8]\***

[1]National Laboratory of Pattern Recognition, Institute of Automation, Chinese Academy of Sciences, Beijing, China; [2]Brainnetome Center, Institute of Automation, Chinese Academy of Sciences, Beijing, China; [3]Department of Neurosurgery, PLA Army General Hospital, Beijing, China; [4]Centre for Hyperbaric Oxygen and Neurorehabilitation, Guangzhou General Hospital of Guangzhou Military Command, Guangzhou, China; [5]Department of Radiology, PLA Army General Hospital, Beijing, China; [6]CAS Center for Excellence in Brain Science and Intelligence Technology, Chinese Academy of Sciences, Beijing, China; [7]Key Laboratory for Neuroinformation of the Ministry of Education, School of Life Science and Technology, University of Electronic Science and Technology of China, Chengdu, China; [8]Queensland Brain Institute, University of Queensland, Brisbane, Australia

**\*For correspondence:**
zjxuruxiang@163.com (RX);
jiangtz@nlpr.ia.ac.cn (TJ)

[†]These authors contributed
equally to this work

**Competing interests:** The
authors declare that no
competing interests exist.

**Reviewing editor:** Klaas Enno
Stephan, University of Zurich and
ETH Zurich, Switzerland

**Abstract** Disorders of consciousness are a heterogeneous mixture of different diseases or injuries. Although some indicators and models have been proposed for prognostication, any single method when used alone carries a high risk of false prediction. This study aimed to develop a multidomain prognostic model that combines resting state functional MRI with three clinical characteristics to predict one year-outcomes at the single-subject level. The model discriminated between patients who would later recover consciousness and those who would not with an accuracy of around 88% on three datasets from two medical centers. It was also able to identify the prognostic importance of different predictors, including brain functions and clinical characteristics. To our knowledge, this is the first reported implementation of a multidomain prognostic model that is based on resting state functional MRI and clinical characteristics in chronic disorders of consciousness, which we suggest is accurate, robust, and interpretable.
DOI: https://doi.org/10.7554/eLife.36173.001

## Introduction

Severe brain injury can lead to disorders of consciousness (DOC). Some patients recover consciousness after an acute brain insult, whereas others tragically fall into chronic DOC. The latter cannot communicate functionally or behave purposefully. Most patients remain bedridden, and require laborious care. The medical community is often confronted with an inability to meet the expectations of the chronic DOC patients' families. The social, economic, and ethical consequences are also tremendous (*Bernat, 2006*). In parallel, although more validations are required, recent pilot studies have proposed new therapeutic interventions, which challenge the existing practice of early treatment discontinuation for a chronic DOC patient (*Schiff et al., 2007*; *Corazzol et al., 2017*; *Yu et al., 2017*). However, before using these novel therapeutic interventions, clinicians first need to determine whether the patient is a suitable candidate. The availability of an accurate and

**eLife digest** Severe brain injury can lead to disorders of consciousness (DOC), such as a coma. Some patients regain consciousness after injury, while others do not. Those who do not recover are unable to communicate or move in purposeful ways, and need long-term care. It can be difficult for physicians to predict which patients will mend. This is mainly based on their observations of the patient's behavior over time. But such perceptions are subjective and vulnerable to errors. More accurate and objective methods are needed.

Several studies suggest that the cause of the injury, the age of the person at the time of injury, and how long the person has had a DOC may predict recovery. Recent studies have shown that using a brain-imaging tool called resting state functional magnetic resonance imaging (fMRI) to measure communication between different parts of the brain may help to calculate the likelihood of recovery.

Now, Song, Yang et al. show that combining resting state fMRI with three pieces of clinical information may help to better predict who will improve. Song et al. created a computer model that forecasts recovery from DOC based on fMRI results, the cause of the person's injury, their age at the time of injury, and how long they have had impaired consciousness. The model could tell which patients would regain consciousness 88% of the time for 112 patients from two medical centers. It also identified several patients who got better despite initial predictions from doctors that they would not.

The experiments show that combining multiple types of information can better predict which patients with DOC will convalesce. Larger studies are needed to confirm that the computer model is reliable. If they do, the model may one day help physicians and families to better plan and manage patients' care.

DOI: https://doi.org/10.7554/eLife.36173.002

robust prognostication is therefore a fundamental concern in the clinical management of chronic DOC patients, as medical treatment, rehabilitation therapy and even ethical decisions depend on this information.

To date, the prognostication for a DOC patient is based on physician observation of the patient's behavior over period that is sufficient to allow determination of whether there is any evidence of awareness. On the one hand, a patient's motor impairment, sensory deficit, cognitive damage, fluctuation of vigilance and medical complications could give rise to misjudgments; on the other hand, for the assessor, a lack of knowledge regarding DOC, poor training and non-use of adequate behavioral scales are additional elements that may contribute to a high possibility of mistakes. Consequently, careful and repeated behavioral assessments are considered to be particularly important for a precise diagnostic and prognostic judgment (*Wannez et al., 2017*). Nonetheless, behavioral assessments are inevitably subjective and vulnerable to a variety of personal interferences (*Giacino et al., 2009*). Physicians and scientists have therefore been seeking accurate and objective markers for diagnosis and prognosis (*Demertzi et al., 2017*; *Noirhomme et al., 2017*).

Several pioneering studies have suggested that the etiology, incidence age and duration of DOC are important indicators for prognosis (*Multi-Society Task Force on PVS, 1994*). Specifically, patients who have non-traumatic brain injury are expected to have a worse functional recovery than traumatic brain injury patients, and young patients were considered more likely to have a favorable outcome than older ones. During the recent decades, some pilot prognostic models have also been explored that are based on features of neurological examination (*Zandbergen et al., 1998*; *Booth et al., 2004*; *Dolce et al., 2008*), abnormalities detected with electroencephalogram (EEG) and evoked potentials (*Steppacher et al., 2013*; *Kang et al., 2014*; *Hofmeijer and van Putten, 2016*; *Chennu et al., 2017*), anatomical and functional changes identified with brain computed tomography (CT), positron emission tomography (PET) and magnetic resonance imaging (MRI) (*Maas et al., 2007*; *Sidaros et al., 2008*; *Neuro Imaging for Coma Emergence and Recovery Consortium et al., 2012*; *Luyt et al., 2012*; *Stender et al., 2014*; *Wu et al., 2015*), and physiological and biochemical disturbances at both the brain and body levels (*Kaneko et al., 2009*; *Rundgren et al., 2009*). Despite many efforts, however, identifying efficient biomarkers for the early

prediction of outcome is still challenging and requires additional research. One of the reasons for this is that the DOC could have many different causes and could be associated with several neuro-pathological processes and different severities, such that any method when used alone carries the risk of false prediction (*Bernat, 2016*; *Rossetti et al., 2016*).

Recently, resting state functional MRI (fMRI) has been widely used to investigate the brain functions of DOC patients. Research suggests that these patients demonstrate multiple changes in brain functional networks, including the default mode (*Vanhaudenhuyse et al., 2010*; *Silva et al., 2015*), executive control (*Demertzi et al., 2014*; *Wu et al., 2015*), salience (*Qin et al., 2015*; *Fischer et al., 2016*), and sensorimotor (*Yao et al., 2015*), auditory (*Demertzi et al., 2015*), visual (*Demertzi et al., 2014a*) and subcortical networks (*He et al., 2015*). The within-network and between-network functional connectivity appeared to be useful indicators of functional brain damage and the likelihood of consciousness recovery (*Silva et al., 2015*; *Di Perri et al., 2016*). Taken together, these studies suggest that the brain networks and functional connectivity detected with resting state fMRI could be valuable biomarkers that can be used to trace the level of consciousness and predict the possibility of recovery.

With advances in medicine, prognostication of a DOC patient has moved toward a multidomain paradigm that combines clinical examination with the application of novel technologies (*Gosseries et al., 2014*). Multidomain assessment has the potential to improve prediction accuracy. More importantly, it can provide reassurances about the importance of each predictor for prognostication by offering concordant evidence (*Stevens and Sutter, 2013*; *Rossetti et al., 2016*). More than 20 years ago, the Multi-Society Task Force on persistent vegetative state (PVS) suggested that the etiology, incidence age and duration of DOC could help to predict the outcome (*Multi-Society Task Force on PVS, 1994*), and numerous studies have subsequently validated the clinical utility of these features (*Jennett, 2005*; *Bruno et al., 2012*; *Estraneo et al., 2013*; *Celesia, 2016*). Therefore, it is possible that a multidomain model that combines these clinical characteristics and resting state fMRI data could improve prognostic predictions at an individual level and could lead to the early identification of patients who could recover consciousness.

The present work had two major objectives. The first aim was to develop an approach to predict the prognosis of an individual DOC patient using clinical characteristics and resting state fMRI. The second aim, building on the first, was to further explore the different prognostic effects of these clinical and brain imaging features.

## Materials and methods

The study paradigm is illustrated in *Figure 1*. Resting state fMRI and clinical data from DOC patients were collected at the so-called $T_0$ time point when the patients' vital signs and consciousness level had stabilized and a diagnosis had been made. Outcomes were assessed at least 12 months after this $T_0$ time point; at a time referred to as the $T_1$ time point. The principal scales included the Coma Recovery Scale Revised (CRS-R) and the Glasgow Outcome Scale (GOS). Instead of predicting diagnosis, this study used the outcome as a target for regression and classification. Using the resting state fMRI and clinical data from the $T_0$ time point in a training dataset, a regression model was first developed to fit each patient's CRS-R score at the $T_1$ time point, after which the optimal cut-off value for classifying individual patients on the basis of consciousness recovery was calculated. In this way, we set up the prognostic regression and classification model. Two independent testing datasets were then used to validate the model.

### Subjects

This study involved three datasets. The datasets referred to as 'Beijing 750' and 'Beijing HDxt' were both collected in the PLA Army General Hospital in Beijing, and the same medical group diagnosed and managed the patients. However, the MRI scanners and imaging acquiring protocols were different for these two datasets: the 'Beijing HDxt' cohort was scanned with a GE signa HDxt 3.0T scanner between May 2012 and December 2013, whereas the 'Beijing 750' cohort was scanned with a GE Discovery MR750 3.0T scanner between January 2014 and May 2016. The dataset referred to as 'Guangzhou HDxt' was collected from the Guangzhou General Hospital of Guangzhou Military Command in Guangzhou, and the MRI data were obtained with a GE signa HDxt 3.0T scanner between April 2011 and December 2014.

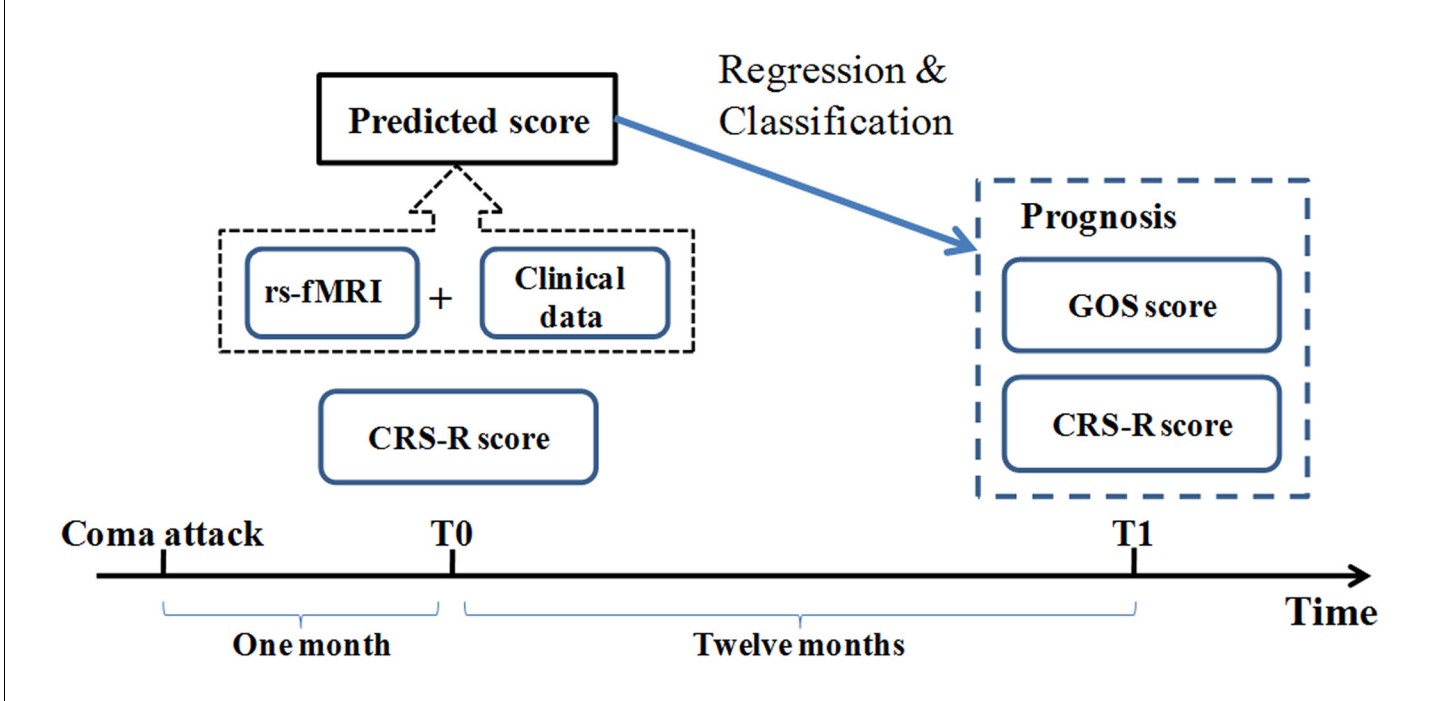

**Figure 1.** Conceptual paradigm of the study. CRS-R: Coma Recovery Scale Revised scale; GOS: Glasgow Outcome Scale.
DOI: https://doi.org/10.7554/eLife.36173.003

The inclusion criterion was that the patients should be at least 1 month after the acute brain insult so that they met the DOC diagnosis. Patients were excluded when there was an unstable level of consciousness (continuous improvement or decline within the two weeks before the $T_0$ time point), uncertain clinical diagnosis (ambiguity or disagreement between examiners), contraindication for MRI or large focal brain damage (>30% of total brain volume).

A total of 160 DOC patients were initially enrolled in this study. Eleven patients were excluded due to large local brain lesions or movement artifacts during MRI scanning. Nine patients died during the period of the follow-up, 16 patients were lost to follow-up, and for 12 patients no definite outcome information was collected at the 12-month endpoint of the follow-up. Thus, according to the inclusion and exclusion criteria and the follow-up results, the 'Beijing 750' dataset included 46 vegetative state/unresponsive wakefulness syndrome (VS/UWS) patients and 17 minimally conscious state (MCS) patients. The 'Beijing HDxt' dataset contained 20 VS/UWS patients and 5 MCS patients, and the 'Guangzhou HDxt' dataset contained 16 VS/UWS patients and 8 MCS patients.

The demographic and clinical characteristics of the patients are summarized in *Table 1*, with additional details provided in *Appendix 1—table 1*, *2* and *3*. The 'Beijing 750' dataset also included 30 healthy participants, and the 'Beijing HDxt' dataset included 10 healthy participants. All of the healthy participants were free of psychiatric or neurological history. These healthy participants are referred to as 'normal controls'. See *Appendix 1—table 4* and *5* for details.

As the 'Beijing 750' dataset involved more patients than the other two datasets, it was used as the training dataset for model development and internal validation, whereas the 'Beijing HDxt' and 'Guangzhou HDxt' datasets were only used for external validation. The study was approved by the Ethics Committee of the PLA Army General Hospital (protocol No: 2011–097) and by the Ethics Committee of the Guangzhou General Hospital of Guangzhou Military Command (protocol No: jz20091287). Informed consent to participate in the study was obtained from the legal surrogates of the patients and from the normal controls.

**Table 1.** Demographic and clinical characteristics of the patients in the three datasets.

| | Beijing_750 (n = 63) | Beijing_HDxt (n = 25) | Guangzhou_HDxt (n = 24) |
|---|---|---|---|
| Gender, M/F | 36/27 | 18/7 | 14/10 |
| Etiology | | | |
| Trauma/Stroke/Anoxia | 17/21/25 | 12/6/7 | 8/0/16 |
| Age at the $T_0$ (years) | | | |
| Mean (SD) | 42.8 (13.8) | 40.7 (15.2) | 39.3 (16.9) |
| Range | 18.0 ~ 71.0 | 18.0 ~ 68.0 | 15.0 ~ 78.0 |
| Time to MRI (months) | | | |
| Range | 1.0 ~ 77.0 | 1.0 ~ 44.0 | 1.0 ~ 10.0 |
| Mean (SD) | 7.4 (12.8) | 5.4 (8.4) | 2.3 (2.4) |
| Median | 3.0 | 3.0 | 1.5 |
| Band | | | |
| [1,3] | 32 | 13 | 20 |
| (3,6] | 15 | 8 | 2 |
| (6,12] | 11 | 3 | 2 |
| >12 | 5 | 1 | 0 |
| Follow-up time (months) | | | |
| Range | 12.0 ~ 51.0 | 14.0 ~ 53.0 | 27.0 ~ 78.0 |
| Mean (SD) | 21.0 (9.8) | 41.7 (8.4) | 52.2 (14.5) |
| Median | 15.0 | 43.0 | 53.0 |
| Band | | | |
| [12,24] | 38 | 2 | 0 |
| (24,48] | 24 | 20 | 8 |
| >48 | 1 | 3 | 16 |
| Diagnosis at $T_0$ | | | |
| MCS/VS | 17/46 | 5/20 | 8/16 |
| CRS-R total score | | | |
| Mean (SD) | 7.3 (2.9) | 6.5 (2.3) | 7.1 (4.1) |
| Range | 3.0 ~ 18.0 | 3.0 ~ 14.0 | 3.0 ~ 17.0 |
| | | | |
| Outcome at $T_1$ | | | |
| CRS-R total score | | | |
| Mean (SD) | 9.9 (5.1) | 12.7 (6.4) | N/A |
| Range | 3.0 ~ 22.0 | 5.0 ~ 23.0 | N/A |
| GOS score | | | |
| GOS = 5 | 0 | 0 | 0 |
| GOS = 4 | 5 | 5 | 1 |
| GOS = 3 | 8 | 7 | 5 |
| GOS <= 2 | 50 | 13 | 18 |

Abbreviations: CRS-R, Coma Recovery Scale–Revised; GOS, Glasgow Outcome Scale; MCS, minimally conscious state; N/A, not available; SD, standard deviation; VS, vegetative state/unresponsive wakefulness syndrome.

DOI: https://doi.org/10.7554/eLife.36173.004

## Clinical measurements

### Diagnosis and consciousness assessments

The diagnosis of each patient in the three datasets was made by experienced physicians according to the CRS-R scale (*Multi-Society Task Force on PVS, 1994*; *Bernat, 2006*; *Magrassi et al., 2016*). In the 'Beijing 750' and 'Beijing HDxt' datasets, the patients underwent the evaluations at least twice weekly within the 2 weeks before the MRI scanning (i.e. the $T_0$ time point). The highest CRS-R score was considered as the diagnosis. The CRS-R includes six subscales that address auditory, visual, motor, oromotor, communication, and arousal functions, which are summed to yield a total score ranging from 0 to 23.

### Outcome assessments

All patients were followed up at least 12 months after MRI scanning, according to the protocols for DOC described in a number of previous studies (*Neuro Imaging for Coma Emergence and Recovery Consortium et al., 2012*; *Luyt et al., 2012*; *Stender et al., 2014*; *Pignat et al., 2016*). Basically, follow-up interviews were performed in four ways, including outpatient visit, assessments by local physicians, home visit, and telephone/video review. Whenever possible, signs of responsiveness were detected or reported, the patient was evaluated either in the unit or at home by the hospital staff. In cases where no change was signaled, patients were examined twice by one hospital physician via telephone/video reviews at the end of the follow-up process.

For the training dataset, 'Beijing 750', two outcome scales were assessed: the GOS and CRS-R. The GOS is one of the most commonly reported global scales for functional outcome in neurology, and provides a measurement of outcome ranging from 1 to 5 (1, dead; 2, vegetative state/minimally conscious state; 3, able to follow commands/unable to live independently; 4, able to live independently/unable to return to work or school; 5, good recovery/able to return to work or school). Although simple to use and highly reliable, the GOS score cannot provide detailed information about individual differences in consciousness level for DOC patients. By contrast, the CRS-R score can assist with prognostic assessment in DOC patients (*Giacino and Kalmar, 2006*). The six subscales in the CRS-R comprise hierarchically arranged items that are associated with brain stem, subcortical and cortical processes. The lowest item on each subscale represents reflexive activity, whereas the highest items represent cognitively mediated behaviors. In order to simplify the modeling, we hypothesized that the higher the total CRS-R score at the follow-up, the better the outcome for the patient. We developed a regression model to fit each patient's CRS-R score at the $T_1$ time point based on their clinical characteristics and resting state fMRI data, and designed a classification model to predict consciousness recovery or not for each patient. The classification accuracy was assessed by comparing the predicted label and the actual GOS score, that is, 'consciousness recovery' (GOS $\geq$ 3) versus 'consciousness non-recovery' (GOS $\leq$ 2).

The testing dataset 'Beijing HDxt' involved both the GOS scores and the CRS-R scores at the $T_1$ time point for each patient. The testing dataset 'Guangzhou HDxt' measured the GOS scores, but not the CRS-R scores at the $T_1$ time point.

## MRI acquisition

All of the participants in the three datasets were scanned with resting state fMRI and $T_1$-weighted 3D high-resolution imaging. During the MRI scanning, the participants did not take any sedative or anesthetic drugs. The resting state fMRI scan was obtained using a $T_2$*-weighted gradient echo sequence, and a high-resolution $T_1$-weighted anatomical scan was obtained to check whether the patients had large brain distortion or focal brain damage. For the training dataset, 'Beijing 750', the resting state fMRI acquisition parameters included TR/TE = 2000/30 ms, flip angle = 90°, axial 39 slices, thickness = 4 mm, no gap, FOV = 240 × 240 mm, matrix = 64 × 64, and 210 volumes (i.e. 7 min). For the testing dataset, 'Beijing HDxt', the resting state fMRI acquisition parameters were as follows: axial 33 slices, TR/TE = 2000/30 ms, flip angle = 90°, thickness = 4 mm, no gap, FOV = 220 × 220 mm, matrix = 64 × 64, and 240 volumes (i.e. 8 min). For the testing dataset, 'Guangzhou HDxt', the resting state fMRI acquisition parameters included axial 35 slices, TR/TE = 2000/30 ms, flip angle = 90°, thickness = 4 mm, no gap, FOV = 240 × 240 mm, matrix = 64 × 64, and 240 volumes (i.e. 8 min).

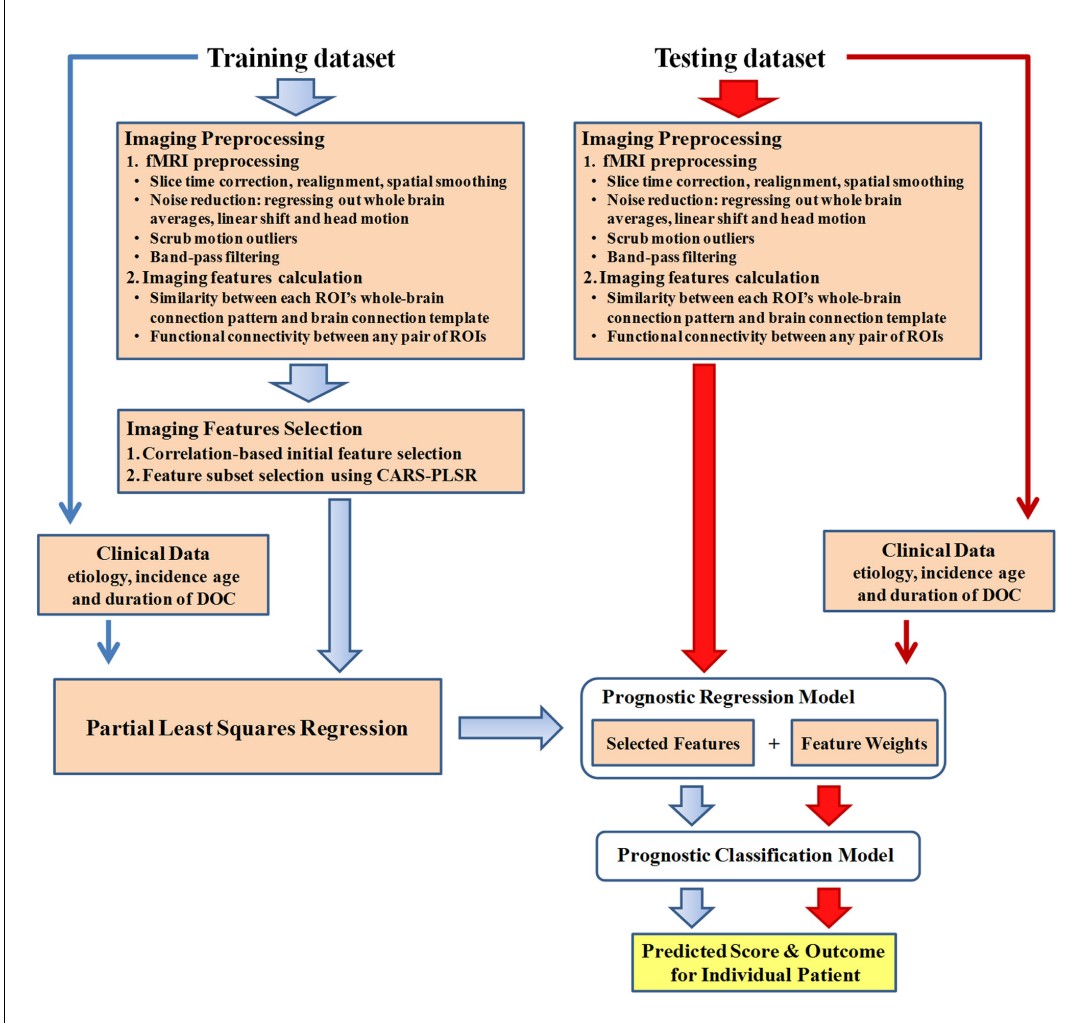

**Figure 2.** Data analysis pipeline. All datasets involved in this study included resting state fMRI and clinical data. For the fMRI data in the training dataset, data analysis first encompassed preprocessing and imaging feature selection and extraction. Partial least square regression was then used to generate the regression model using the selected imaging features and clinical features in the training dataset. In this way, a prediction score that depicts the possibility of consciousness recovery was computed for each patient. The optimal cut-off value for classifying an individual patient as responsive or non-responsive was then calculated, and the prognostic classification model was obtained. The two testing datasets were only used to validate externally the regression and classification model.

DOI: https://doi.org/10.7554/eLife.36173.005

## Data analysis

The data analysis pipeline is illustrated in *Figure 2*.

## Imaging preprocessing

Preprocessing and connectivity calculation were performed in the same way for the training dataset and the two testing datasets. All resting-state fMRI scans were preprocessed using SPM8 (SPM, RRID:SCR_007037) and in-house Matlab codes. Specifically, the first five volumes of each subject were discarded. The remaining resting-state fMRI volumes were corrected for slice timing differences and realigned to the first volume to correct for inter-scan movements. The functional images were then spatially smoothed with a Gaussian kernel of $6 \times 6 \times 6$ mm full-width at half maximum. Linear regression was used to remove the influence of head motion, whole brain signals and linear trends. The variables regressed out included 12 motion parameters (roll, pitch, yaw, translation in three dimensions and their first derivatives), the average series of the signals within the brain, and the regressors for linear trends.

Motion artifact is increasingly recognized as an important potential confound in resting state fMRI studies. Any particular motion may produce a wide variety of signal changes in the fMRI data, and may thus introduce complicated shifts in functional connectivity analysis. This problem was particularly serious for the DOC patients, as they were unlikely to follow the experimental instructions and control their head motion. To balance the demands of noise reduction and data preservation, we censored volumes that preceded or followed any movement (framewise displacement, FD) greater than 1.5 mm. The FD is a summarization of the absolute values of the derivatives of the translational and rotational realignment estimates (after converting the rotational estimates to displacement at 50 mm radius) (*Power et al., 2015*). The head motion measures were achieved in the preprocessing step of realignment using SPM. To obtain reliable Pearson's correlations for functional connectivity, the patients with less than 50 volumes worth of remaining data were excluded. More information about the analysis and validation of controls for motion-related artifacts are provided in Appendix 4.

Finally, to reduce low-frequency drift and high-frequency noise, band-pass filtering (0.01–0.08 Hz) was only performed on volumes that survived motion censoring.

## Definition of networks and regions of interest

As noted in the introduction, multiple functional brain networks are disrupted in DOC patients. Among these impaired networks, six (the default mode, executive control, salience, sensorimotor, auditory, and visual networks) show system-level damages and significant correlations with behavioral assessments (*Demertzi et al., 2014, 2015*). We therefore defined a total of 22 regions of interest (ROIs) to probe these six brain networks. The definitions of the 22 ROIs were based on the results of a series of previous brain functional studies (*Seeley et al., 2007*; *Raichle, 2011*; *Demertzi et al., 2015*), and their names and Montreal Neurological Institute (MNI) coordinates are listed in Appendix 2.

The connection templates of the six brain networks were first investigated within the normal control group. In addition to the above-mentioned preprocessing stages, the resting state fMRI scans of the normal controls in the training dataset were transformed into MNI standard space. For each of the six networks, time series from the voxels contained in the various ROIs were extracted and averaged together. The averaged time series were then used to estimate whole-brain correlation r maps that were subsequently converted into normally distributed Fisher's z-transformed correlation maps. Group functional connectivity maps for each of the six networks were then created with a one-sample t test (see Appendix 3 for details). Notably, the T map included both positive and negative values. We used the six T maps as the brain connection templates of the corresponding brain networks in the healthy population, which would assist to define one type of imaging features, that is the connection feature of the ROI. More information about the connection features of the ROIs are provided in the following section.

The conventional fMRI preprocess normalizes individual fMRI images into a standard space defined by a specific template image. Our goal was to extend this conventional approach to generate a functional connectivity image for each patient in his/her own imaging space. During the preprocessing of each patient's fMRI scans, the 22 ROIs and six brain connection templates were therefore spatially warped to individual fMRI space and resampled to the voxel size of the individual fMRI image. We also developed tools to check the registration for each subject visually, some examples of which are provided in Appendix 5 and *Supplementary file 1*.

## Calculation of imaging features

We designed two types of imaging features from the resting state fMRI, one being the functional connectivity between each pair of 22 ROIs, and the other being the spatial resemblance between the functional connection patterns of each ROI and the brain connection templates across the whole brain. The functional connectivity was based on the Pearson's correlation coefficients, while the spatial resemblance was conceptually similar to the template-matching procedure (*Greicius et al., 2004*; *Seeley et al., 2007*; *Vanhaudenhuyse et al., 2010*). The basis of template matching is that the greater the spatial consistency that exists between the template of a brain network and a specific connectivity map (for example, a component in an independent component analysis), the stronger the possibility that the connectivity map belongs to that brain network. Here, for each ROI of an individual DOC patient, we first computed the Pearson's correlation coefficients between the time-

course of the ROI and that of each voxel within the brain so as to obtain a functional connectivity map, and subsequently converted the functional connectivity map to a normally distributed Fisher's z transformed correlation map. Next, we calculated the Pearson's correlation coefficients between the Fisher's z transformed correlation map and the corresponding brain connection template wrapped to individual fMRI space across each voxel within the brain. A greater correlation coefficient between the two maps suggests that there is more spatial resemblance between the functional connectivity map of the ROI and the normal brain connection template. Our assumption was that the more spatial consistency that existed between the connectivity map of the ROI in a DOC patient and the brain connection template, the more intact the corresponding brain function of the ROI in this individual. In this way, we defined the connection feature of the ROI with the spatial resemblance.

Overall, for each participant in this study, there were 231 ($22 \times 21/2$) functional connectivity features and 22 brain area connection features.

## Imaging feature selection

Feature selection techniques have been widely adopted in brain analysis studies, in order to produce a small number of features for efficient classification or regression, and to reduce overfitting and increase the generalization performance of the model (*Fan et al., 2007*; *Dosenbach et al., 2010*; *Drysdale et al., 2017*). Feature ranking and feature subset selection are two typical feature selection methods (*Guyon and Elisseeff, 2003*). Feature subset selection methods are generally time consuming, and even inapplicable when the number of features is extremely large, whereas ranking-based feature selection methods are subject to local optima. Therefore, these two feature selection methods are usually used jointly. Here, we first used a correlation-based feature selection technique to select an initial set of features, and then adopted a feature subset selection method for further selection.

As a univariate method, correlation-based feature selection is simple to run and understand, and measures the linear correlation between each feature and the response variable. Here, the image features (i.e. functional connectivity features and brain area connection features) that significantly correlated to the CRS-R scores at the $T_1$ time point across the DOC patients in the training dataset were retained for further analysis.

Competitive adaptive reweighted sampling coupled with partial least squares regression (CARS-PLSR, http://libpls.net/) was then used for further feature subset selection (*Li et al., 2009, 2014*). Briefly, CARS-PLSR is a sampling-based feature selection method that selects the key informative variables by optimizing the model's performance. As it provides the influence of each variable without considering the influence of the remainder of the variables, CARS-PLSR is efficient and fast in carrying out feature selection (*Mehmood et al., 2012*), and has therefore been used to explore possible biomarkers in medicine (*Tan et al., 2010*) and for wavelength selection in chemistry (*Fan et al., 2011*). Using CARS-PLSR, we selected a subset of key informative imaging features.

Notably, both the correlation-based and CARS-PLSR feature selection methods filtered the features from the original feature set without any transformations. This made the prognostic regression model easier to interpret, as the imaging predictors were associated with either brain regions or functional connectivity.

## Prognostic modeling and assessments of predictor importance

PLSR is able to handle multicollinearity among the predictors well (*Wold et al., 2001*; *Krishnan et al., 2011*). It was therefore used to generate the prognostic regression model in the training dataset 'Beijing 750'. Given that clinical characteristics—including the etiology, incidence age and duration of DOC—have been verified as useful prognostic indicators, we designated the selected imaging features and the three clinical characteristics at the $T_0$ time point as independent co-variates and the CRS-R score at the $T_1$ time point as the dependent variable. Among the three clinical characteristics, the incidence age and duration of DOC were quantitative variables, whereas the etiology was a qualitative variable. In accordance with a previous study (*Estraneo et al., 2010*), we categorized the etiology into three types: traumatic brain injury, stroke and anoxic brain injury. Thus, two dummy variables for etiology were designed and included in the model. Prior to model training, all involved predictors were centered and normalized (i.e. transformed into Z-scores). The prognostic regression model therefore took the imaging and clinical features as input and returned

a predicted score as output. In the training dataset 'Beijing 750', we used cross-validation to decide that the number of latent variables for PLSR was three. To evaluate the regression model, the coefficient of determination $R^2$ between the predicted scores and the CRS-R scores at the $T_1$ time point was calculated, and the Bland-Altman plot was used to measure the agreement between them.

Next, receiver operating characteristic (ROC) curves were plotted for the predicted scores. The optimal cut-off value for classifying an individual patient as having recovered consciousness or not was appointed to the point with the maximal sum of true positive and false negative rates on the ROC curve. Individual patients were classified as exhibiting recovery of consciousness if their predicted scores were higher than or equal to the cut-off value, otherwise as consciousness non-recovery. The classification accuracy was calculated by comparing the predicted label and the actual GOS score, that is 'consciousness recovery' (GOS $\geq$ 3) versus 'consciousness non-recovery' (GOS $\leq$ 2).

As model interpretation is an important task in most applications of PLSR, there has been considerable progress in the search for optimal interpretation methods (*Kvalheim and Karstang, 1989*; *Kvalheim et al., 2014*). In this study, using the Significant Multivariate Correlation (sMC) method (*Tran et al., 2014*), we assessed predictor importance in the prognostic regression model. The key points in sMC are to estimate the correct sources of variability resulting from PLSR (i.e. regression variance and residual variance) for each predictor, and use them to determine statistically a variable's importance with respect to the regression model. The F-test values (termed the sMC F-values) were used to evaluate the predictors' importance in the prognostic regression model.

## Internal validation of model

The prognostic regression model was internally validated using bootstrap sampling (*Steyerberg, 2008*). Specifically, bootstrap samples were drawn with replacement from the training dataset 'Beijing 750' such that each bootstrap sampling set had a number of observations equal to that of the training dataset. Using a bootstrap sampling set, correlation-based feature selection and CARS-PLSR were first used to select the feature subset, after which the PLSR was used to generate a prognostic model. We then applied the model to the bootstrap sampling set and the original training dataset, and calculated the coefficient of determination $R^2$ of each of the two datasets. The difference between the two coefficients of determination was defined as the optimism. This process was repeated 1000 times to obtain a stable estimate of the optimism. Finally, we subtracted the optimism estimate from the coefficient of determination $R^2$ of the 'Beijing 750' training dataset to obtain the optimism-corrected performance estimate.

In addition, out-of-bag (OOB) estimation was used as an estimate of model classification performance in the training dataset (*James et al., 2013*). Specifically, for the original training dataset $x$, we left out one sample at a time and denoted the resulting sets by $x_{(-1)}, ..., x_{(n)}$. From each leave-one-out set $x_{(-i)}$, 1000 bootstrap learning sets of size $n-1$ were drawn. On every bootstrap learning set generated from $x_{(-i)}$, we carried out feature selection, built a PLSR regression and classification model, and applied the model to the test observation $x_i$. A majority vote was then made to give a class prediction for observation $x_i$. Finally, we calculated the accuracy for the whole training dataset $x$.

## External model validation

External validation is essential to support the general applicability of a prediction model. We ensured external validity by testing the model in two testing datasets, neither of which included samples that were considered during the development of the model. First, using the prognostic regression model, we calculated one predicted score for each patient in the two testing datasets. As the 'Beijing HDxt' dataset assessed the patients' CRS-R scores at the $T_1$ time point, we calculated the coefficient of determination $R^2$ between the predicted scores and the patients' CRS-R scores at this time point. The Bland-Altman plot was also determined. Finally, the patients in the two testing datasets were assessed as achieving consciousness recovery or not on the basis of the cut-off threshold obtained using the training dataset. The performance of the classification, including the accuracy, sensitivity and specificity, was determined.

## Comparison between single-domain model and combination model

Using the modeling and validation method described above, we examined the predictability and generalizability in the two testing datasets on the basis of the clinical features alone or the imaging features alone.

In addition, to compare the two types of single-domain models and the combination model, we used bootstrap resampling to obtain the distribution of the prediction accuracies in the two testing datasets based on each of the three types of models. We first resampled with replacement from the training dataset, and built a regression and classification model based on the clinical features alone, the neuroimaging features alone, or the combination of the two-domain features. We then tested the classification accuracy in the two testing datasets using the three types of models. In this way, we obtained the distribution of the prediction accuracies using each of the three types of models. Next, we used repeated measures ANOVA to determine whether or not the performances of the three types of models were the same; we also used $\Psi$, the root-mean-square standardized effect, to report the effect sizes of the mean differences between them.

## Comparison between the proposed modeling method and linear SVM

We compared the prediction performances between the proposed modeling method and linear SVM. The code for SVM was downloaded from LIBSVM (LIBSVM, RRID:SCR_010243). The 253 imaging features and the four clinical features were integrated into one feature vector. No feature selection was adopted in the linear SVM-based classification. The patients with GOS $\geq 3$ were labeled as 1, with the others being designated as $-1$ (i.e. GOS $\leq 2$).

Similarly, the OOB estimation process was used to estimate the performance of linear SVM in the training dataset 'Beijing 750'. Next, using all the samples in the training dataset 'Beijing 750', we trained a linear SVM-based classifier and then tested the predictive accuracy in the two testing datasets.

## Results

### Imaging feature selection

#### Correlation-based feature selection

Using the training dataset, we found that some imaging features significantly correlated to the CRS-R scores at the $T_1$ time point. For example, the connection features of some brain areas, including the anterior medial prefrontal cortex (aMPFC), posterior cingulate cortex/precuneus (PCC) and right lateral parietal cortex in the default mode network, and the dorsal medial prefrontal cortex (DMPFC) and left lateral superior parietal cortex in the executive control network, displayed significant correlations to the CRS-R $T_1$ scores across the DOC patients. We also found numerous examples of significant correlation between functional connectivity and the CRS-R score at the $T_1$ time point, with these functional connectivities being distributed both within and between brain networks. More information about the correlations between the imaging features and the CRS-R scores at the $T_1$ time point are provided in Appendix 6.

#### CARS-PLSR feature selection

*Figure 3* shows the final imaging features selected with CARS-PLSR. Specifically, the brain area connection features included the aMPFC and PCC in the default mode network, and the DMPFC in the executive control network. The functional connectivity features included the connectivity between the aMPFC in the default mode network and the DMPFC in the executive control network, as well as between the middle cingulate cortex in the auditory network and the right lateral primary visual cortex in the visual network. More information about the feature selection by bootstrapping is provided in Appendix 7.

### Prognostic regression model and predictor importance

The prognostic regression model is presented in *Figure 4*. On the basis of the regression formula, we noted some interesting findings. First, there were both positive and negative weights. In particular, the weights were all positive for the three brain area connection features, whereas the weight for

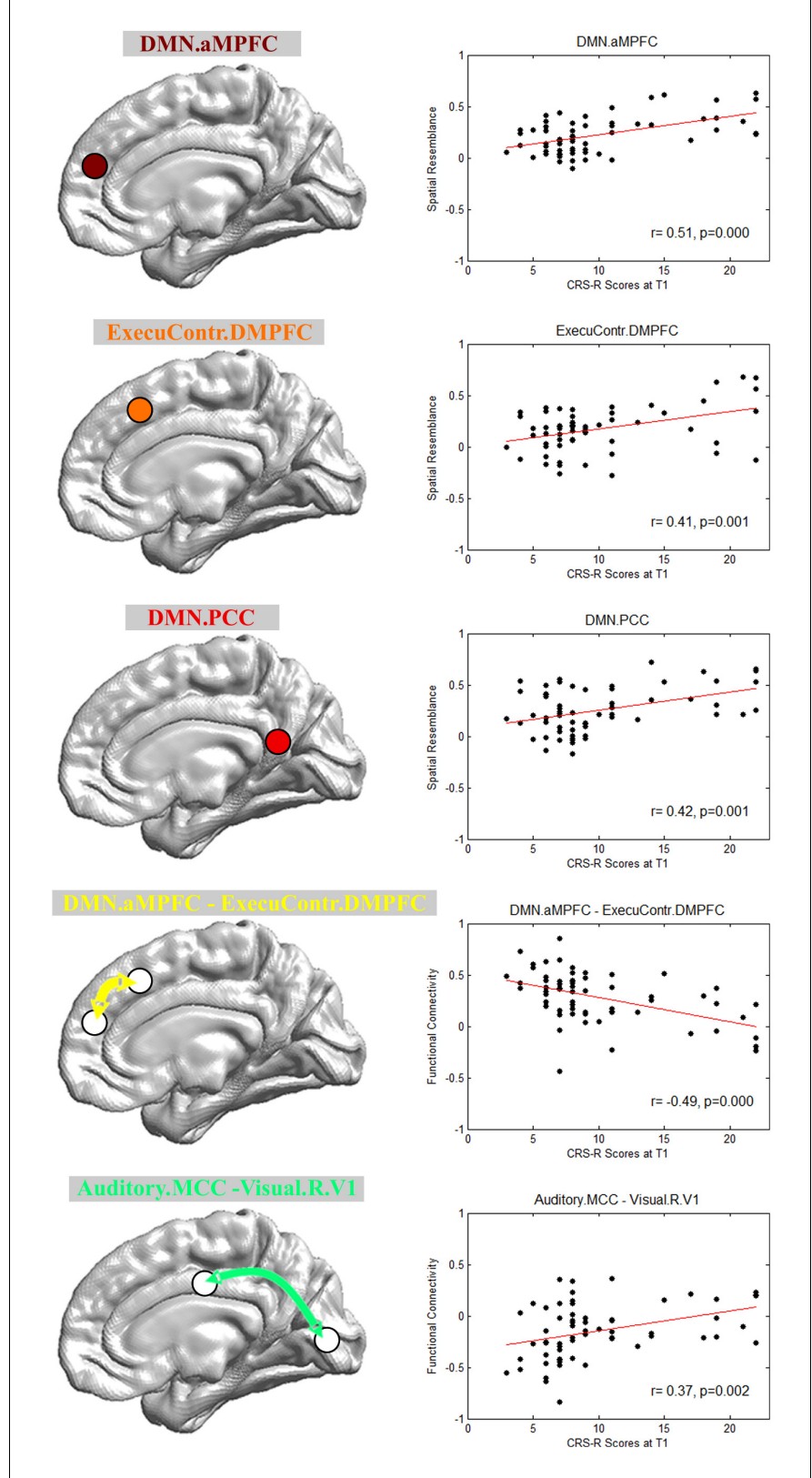

**Figure 3.** Imaging features involved in the prognostic regression model. DMN.aMPFC, anterior medial prefrontal cortex in the default mode network; DMN.PCC, posterior cingulate cortex/precuneus in the default mode network; ExecuContr.DMPFC, dorsal medial prefrontal cortex in the executive control network; Auditory.MCC, middle

*Figure 3 continued*

cingulate cortex in the auditory network; Visual.R.V1, right lateral primary visual cortex in the visual network. DMN. aMPFC—ExecuContr.DMPFC: the functional connectivity between DMN.aMPFC and ExecuContr.DMPFC; Auditory.MCC—Visual.R.V1: the functional connectivity between Auditory.MCC and Visual.R.V1.

DOI: https://doi.org/10.7554/eLife.36173.006

the functional connectivity feature between the aMPFC in the default mode network and the DMPFC in the executive control network was negative. Interestingly, this connection had the maximum sMC F-value as shown in *Figure 4B*. In addition, the age and the anoxic etiology had negative weights, and the age predictor had the largest sMC F-value among the four clinical features.

## Prognostic classification model and internal validation

*Figure 5A* presents the predicted score for each patient in the training dataset. As shown in *Figure 5B*, there was good agreement between the CRS-R scores at the $T_1$ time point and the predicted scores. The apparent coefficient of determination $R^2$ was equal to 0.65 (permutation test, p=0.001), and the Bland-Altman plot verified the consistency between the predicted and achieved scores (one sample T test, p = 1.0). The prognostic regression model was internally validated using bootstrapping. The optimism-corrected coefficient of determination $R^2$ was equal to 0.28.

*Figure 5C* illustrates the number and proportion of DOC patients in different bands of predicted scores. We found that the proportion of the patients with a 'consciousness recovery' outcome in the patient cohorts rose in conjunction with an increase in the predicted score. The higher the predicted

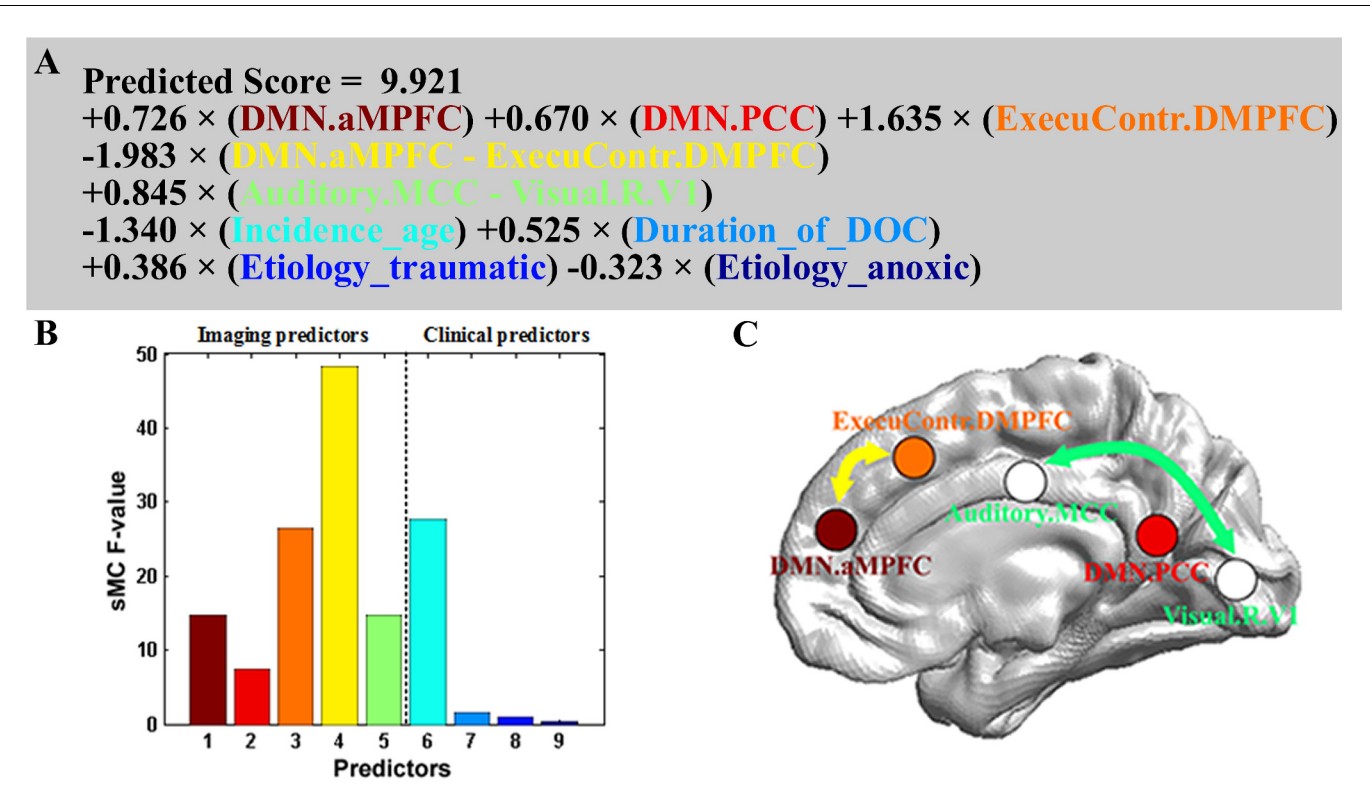

**Figure 4.** Prognostic regression model. In the three subplots, each color denotes a particular predictor. (**A**) Regression formula. (**B**) Predictor importance for each predictor in prognostic regression model. The vertical axis represents the sMC F-test value. The larger the sMC F-value, the more informative the predictor with respect to the regression model. (**C**) The imaging features in the model are rendered on a 3D surface plot template in medial view.

DOI: https://doi.org/10.7554/eLife.36173.007

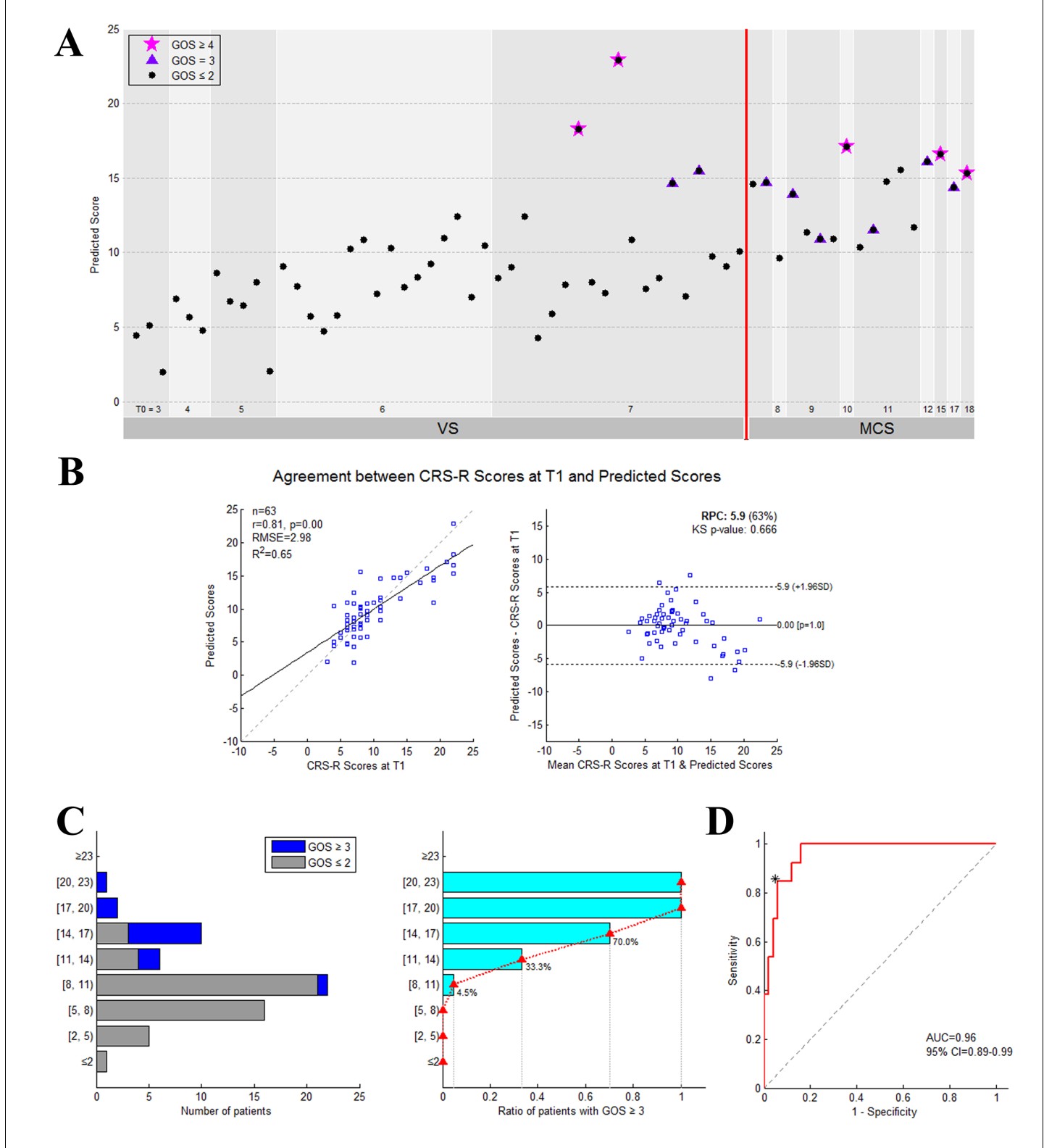

**Figure 5.** The performance of the prediction model on the training dataset. (**A**) Individual predicted scores for each DOC patient in the training dataset. The CRS-R score at the $T_0$ time point is shown on the x axis and the predicted score on the y axis. The patients diagnosed as VS/UWS at the $T_0$ time point are shown to the left of the vertical red solid line, whereas the patients diagnosed as MCS at this time point are shown to the right. The purplish red pentagram, imperial purple triangle and blank circle mark the patients with a GOS score $\geq 4$, $=3$ and $\leq 2$, respectively, at the $T_1$ time point.

*Figure 5 continued on next page*

Figure 5 continued

(B) Agreement between the CRS-R scores at the $T_1$ time point and the predicted scores. The left panel shows the correlation between the CRS-R scores at the $T_1$ time point and the predicted scores, and the right panel shows the differences between them using the Bland-Altman plot. (C) Bar chart showing the numbers or proportions of DOC patients in each band of predicted scores. In these two panels, the y axis shows the predicted score. (D) The area under the receiver-operating characteristic (ROC) curve. The star on the curve represents the point with the maximal sum of true positive and false negative rates on the ROC curve, which were chosen as the cut-off threshold for classification. Here, the corresponding predicted score = 13.9.

DOI: https://doi.org/10.7554/eLife.36173.008

score, the higher the proportion of patients who exhibited a favorable outcome. *Figure 5D* shows the area under the ROC curve (AUC = 0.96, 95% CI = 0.89–0.99). On the basis of the ROC curve for the training dataset, a threshold 13.9 was selected as the cut-off point to classify the recovery of individual patients. In other words, if the predicted score for a patient was equal to or larger than 13.9, the classification model designated the label 'consciousness recovery' for this patient, otherwise 'consciousness non-recovery'. The classification accuracy was assessed by comparing the predicted and actual outcomes, that is 'consciousness recovery' (GOS $\geq$ 3) versus ' consciousness non-recovery' (GOS $\leq$ 2). Using this method, the classification accuracy in the training dataset was up to 92%. Specifically, the sensitivity was 85%, the specificity was 94%, the positive predictive value (PPV) was 79%, the negative predictive value (NPV) was 96%, and the $F_1$ score was 0.81.

The OOB was able to provide the mean prediction error on each training sample and to estimate the generalizability of our method in the training dataset. Using the OOB estimation, we found that the prediction accuracy in the training dataset 'Beijing 750' was 89%, and the sensitivity, specificity, PPV and NPV were 69%, 94%, 100%, and 87%, respectively.

## External validation of the model

The performance of the prediction model on the two testing datasets is illustrated in *Figure 6*. As we assessed the CRS-R scores at the $T_1$ time point for the patients in the 'Beijing HDxt' dataset, we calculated the coefficient of determination $R^2$ between these scores and the predicted scores. The $R^2$ was equal to 0.35 (permutation test, p=0.005), with the Bland-Altman plot verifying the consistency between the predicted and actual scores (one sample T test, p=0.89). Using the predicted score 13.9 as the threshold, we then tested the classification accuracy on the two testing datasets. We found that, for the 'Beijing HDxt' dataset, the prediction accuracy was up to 88% (sensitivity: 83%, specificity: 92%, PPV: 92%, NPV:86%, $F_1$ score: 0.87; permutation test, p<0.001), while for the 'Guangzhou HDxt' dataset it was also up to 88% (sensitivity: 100%, specificity: 83%, PPV: 67%, NPV: 100%, $F_1$ score: 0.80; permutation test, p<0.001). Notably, our model demonstrated good sensitivity and specificity for both the 'subacute' patients (i.e. duration of unconsciousness $\leq$3 months) and those in the chronic phase (i.e. duration of unconsciousness >3 months), as shown in *Figure 7*. More interestingly, for the testing dataset 'Beijing HDxt', eight DOC patients who were initially diagnosed as VS/UWS subsequently recovered consciousness. Using the proposed model, we could successfully identify seven of these and there was only one false-positive case. That is, for the VS/UWS patients, the model achieved 90.0% accuracy (sensitivity: 87.5%, specificity: 91.7%, PPV: 87.5%, NPV: 91.7%, $F_1$ score: 0.875).

To test robustness, we evaluated whether the present prognostic regression model generalized to the healthy subjects scanned in the 'Beijing 750' training dataset (n = 30) and to the 'Beijing HDxt' testing dataset (n = 10). We found that both the healthy subjects and the 'consciousness recovery' patients had significantly higher predicted imaging subscores than the 'consciousness non-recovery' patients (two-sample T test, p<0.05). Additional information is provided in Appendix 8.

## Comparison of the single-domain and combination models

When only the clinical features were used to build the predictive model, the prediction accuracy for the 'Beijing HDxt' dataset was 68% (sensitivity: 58%, specificity: 77%, PPV: 70%, NPV: 67%, $F_1$ score: 0.64), whereas for the 'Guangzhou HDxt' dataset, it was 83% (sensitivity: 100%, specificity: 78%, PPV: 60%, NPV: 100%, $F_1$ score: 0.75). When only the imaging features were involved in the model, the prediction accuracy for the 'Beijing HDxt' dataset was 80% (sensitivity: 67%,

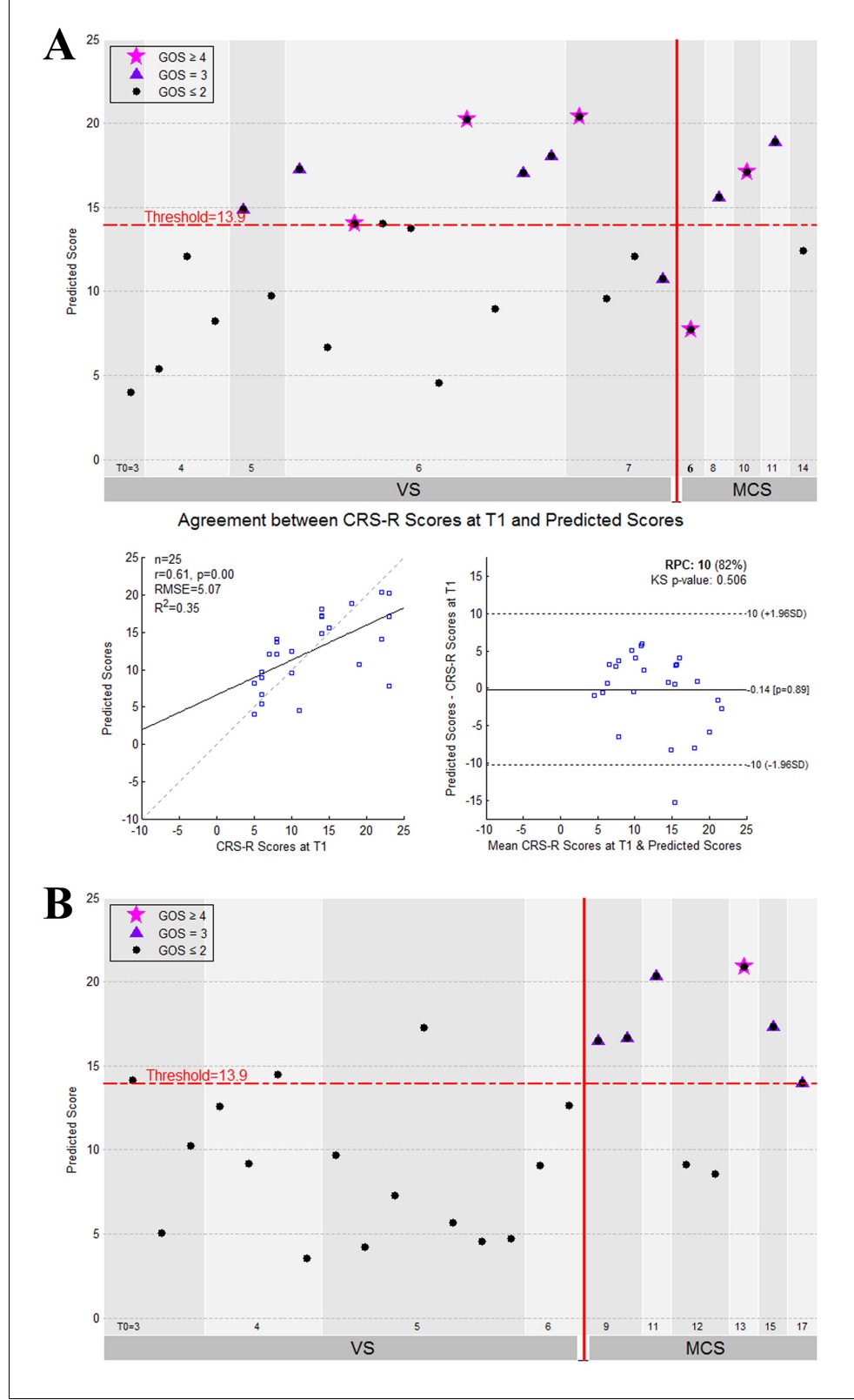

**Figure 6.** The performance of the prediction model on the two testing datasets. (**A**) The individual predicted score (top panel) and agreement between the CRS-R scores at the $T_1$ time point and the predicted scores (bottom panel) for the testing dataset 'Beijing HDxt'. (**B**) The individual predicted score for each DOC patient in the testing dataset 'Guangzhou HDxt'. The legend description is the same as for **Figure 5**.

DOI: https://doi.org/10.7554/eLife.36173.009

specificity: 92%, PPV: 89%, NPV: 75%, $F_1$ score: 0.76), whereas for the 'Guangzhou HDxt' dataset, it was 79% (sensitivity: 100%, specificity: 72%, PPV: 55%, NPV: 100%, $F_1$ score: 0.71).

Using bootstrapping, we obtained the distribution of the prediction accuracies in the two testing datasets with each of the three types of models. In the 'Beijing HDxt' testing dataset, the

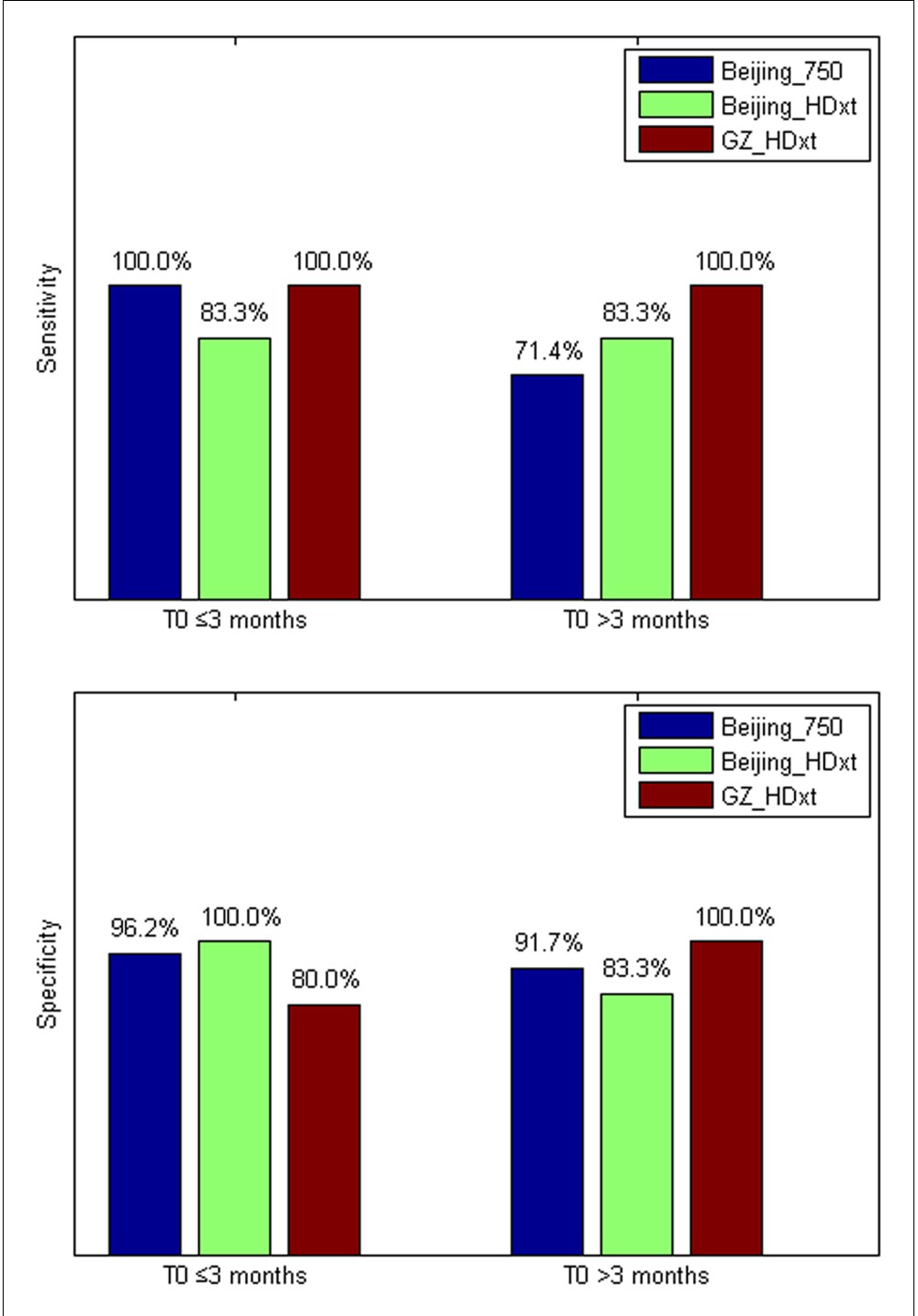

**Figure 7.** The sensitivity and specificity in the 'subacute' patients (i.e. duration of unconsciousness $T_0 \leq 3$ months) and those in the chronic phase (i.e. duration of unconsciousness $T_0 > 3$ months), respectively.
DOI: https://doi.org/10.7554/eLife.36173.010

means±standard deviations of the distribution of the prediction accuracies were 0.815±0.050, 0.811±0.044, and 0.666±0.037 for the combination model, the model using imaging features alone, and the model using clinical features alone, respectively. We found that there were significant differences between the means of the classification accuracies using the three types of models (repeated measures ANOVA, p<0.001). Subsequently, we conducted pairwise comparisons. We found that there was significant difference between the combination model and the model s using the imaging feature alone (paired sample t-test, p=0.001) or using the clinical feature alone (paired sample t-test, p<0.001). We also found that there was significant difference between the model using the imaging feature alone and the model using the clinical feature alone (paired sample t-test, p<0.001). Using effect-size analysis, we found that there was a mean difference of $\Psi=0.004$ (95% CI = [0.002, 0.007]) between the combination method and the method using only imaging features, and $\Psi=0.149$ (95% CI = [0.147, 0.152]) between the combination method and the method using only clinical features. We also observed a mean difference of $\Psi = 0.145$ (95% CI = [0.142, 0.147]) between the methods using only imaging features and only clinical features.

In the 'Guangzhou HDxt' testing dataset, the mean±standard deviation of the distribution of the prediction accuracies was 0.863±0.051, 0.783±0.044, and 0.829±0.086 for the combination model, the model using imaging features alone, and the model using clinical features alone, respectively. Similarly, we found that there were significant differences between the mean of the classification accuracies using the three types of models (repeated measures ANOVA, p<0.001), and there was significant difference between the combination model and the models using imaging features alone (paired sample t-test, p<0.001) or using clinical features alone (paired sample t-test, p<0.001). Using effect-size analysis, we found a mean difference of $\Psi = 0.080$ (95% CI = [0.076, 0.084]) between the combination model and the model using the imaging features alone, and $\Psi = 0.034$ (95% CI = [0.028, 0.040]) between the combination model and the model using only clinical features. We also observed a mean difference of $\Psi = -0.046$ (95% CI = [–0.053, –0.040]) between the model using imaging features alone and that using only clinical features.

Therefore, in both testing datasets, the combination of imaging and clinical features demonstrated higher accuracy than the use of the single domain features alone. In addition, use of the imaging features alone had higher predictive power in comparison to use of the clinical features alone in the 'Beijing HDxt' dataset, whereas the opposite condition was observed in the 'Guangzhou HDxt' dataset, suggesting that the two testing datasets might be heterogeneous. More information about the single-domain models are provided in *Supplementary file 2*.

## Comparison between the proposed modeling method and linear SVM

Using the OOB estimation, we found that the accuracy of the linear SVM-based classification method in the training dataset 'Beijing 750' was 83% (sensitivity: 31%, specificity: 96%, PPV: 100%, NPV: 81%), which was lower than the accuracy of our proposed modeling method (i.e. accuracy: 89%, sensitivity: 69%, specificity: 94%, PPV: 100%, NPV: 87%). On the other hand, the linear SVM-based classification method achieved an accuracy of 76% (sensitivity: 58%, specificity: 92%, PPV: 88%, NPV: 71%) and 88% (sensitivity: 100%, specificity: 83%, PPV: 67%, NPV: 100%) in the 'Beijing HDxt' testing dataset and the 'Guangzhou HDxt' testing dataset, respectively. That is, the accuracy in the 'Beijing HDxt' testing dataset was lower than that in our method, whereas the accuracy in the 'Guangzhou HDxt' testing dataset was similar to that of our approach. Therefore, taking together the performance comparisons in both the training dataset and the two testing datasets, we believe that our method based on feature selection and PLSR should have higher prediction accuracy and better generalizability in comparison to linear SVM.

## Discussion

In this paper, we describe the development of a prognostic model that combines resting state fMRI with three clinical characteristics to predict one-year outcomes at the single-subject level. The model discriminated between patients who would later recover consciousness and those who would not with an accuracy of around 88% in three datasets from two medical centers. It was also able to identify the prognostic importance of different predictors, including brain functions and clinical characteristics. To our knowledge, this is the first reported implementation of a multidomain prognostic model that is based on resting state fMRI and clinical characteristics in chronic DOC. We therefore

suggest that this novel prognostic model is accurate, robust, and interpretable. For research only, we share the prognostic model and its Matlab code at a public download resource (https://github.com/realmsong504/pDOC; copy archived at https://github.com/elifesciences-publications/pDOC).

Brain functions are mediated by the interactions between neurons within different neural circuits and brain regions. Functional imaging can detect the local activity of different brain regions and explore the interactions between them, and has demonstrated potential for informing diagnosis and prognosis in DOC. On the one hand, many studies have focused on one modality of brain functional imaging, such as PET (*Phillips et al., 2011*), SPECT (*Nayak and Mahapatra, 2011*), task fMRI (*Owen et al., 2006*; *Coyle et al., 2015*), or resting state fMRI (*Demertzi et al., 2015*; *Qin et al., 2015*; *Wu et al., 2015*; *Roquet et al., 2016*). On the other hand, some cross-modality studies have compared the diagnostic precision between imaging modalities, for example, comparing PET imaging with task fMRI (*Stender et al., 2014*) or comparing PET, EEG and resting state fMRI (*Golkowski et al., 2017*). In our study, by combining brain functional networks detected from resting state fMRI with three clinical characteristics, we built a computational model that allowed us to make predictions regarding the prognosis of DOC patients at an individual level. We compared the models separately using only the imaging features or only the clinical characteristics and found that the combination of these predictors achieved greater accuracy. This validated the need for the use of accumulative evidence stemming from multiple assessments, each of which has different sensitivity and specificity in detecting the capacity for recovery of consciousness (*Demertzi et al., 2017*). Validations in additional and unseen datasets were also undertaken to evaluate the feasibility of the predictive model. Our results showed about 88% average accuracy across the two testing datasets, and good sensitivity and specificity in both the 'subacute' patients (i.e. 1 months $\leq$ duration of unconsciousness $\leq$ 3 months) and those in the prolonged phase (i.e. duration of unconsciousness >3 months), which suggested good robustness and the generalizability of our model.

Further, the sensitivity of 83% and 100% obtained across the two testing datasets demonstrated a low false-negative rate, which would avoid predicting non-recovery in a patient who can actually recover. Our method successfully identified 16 out of the total 18 patients who later recovered consciousness in the two testing datasets. In parallel, the specificity across the two testing datasets was up to 92% and 83%, respectively. Taken together, these results suggest that our method can precisely identify those patients with a high-potential for recovery consciousness and concurrently reduce false positives in predicting low-potential patients. In addition, although it has been widely considered that the prospect of a clinically meaningful recovery is unrealistic for prolonged DOC patients (*Wijdicks and Cranford, 2005*), our model correctly predicted 9 out of 10 DOC patients with longer than or equal to three months duration of DOC who subsequently recovered consciousness, including three patients with DOC longer or equal to 6 months duration, suggesting that it can also be applied to prolonged DOC patients.

According to the surviving consciousness level, DOC patients can be classified into distinct diagnostic entities, including VS/UWS and MCS. As MCS is often viewed as a state with a potentially more favorable outcome (*Luauté et al., 2010*), a misdiagnosis of VS/UWS could heavily bias the judgment of the prognosis, the medical treatment options and even the associated ethical decisions. Some influential studies have found that a few VS/UWS patients exhibit near-normal high-level cortical activation in response to certain stimuli or commands (*Owen et al., 2006*; *Monti et al., 2010*), and that late recovery of responsiveness and consciousness is not exceptional in patients with VS/UWS (*Estraneo et al., 2010*). Instead of predicting diagnosis, this study used one-year outcome as a target for regression and classification. Our method, which is based on the combined use of clinical and neuroimaging data, successfully identified seven out of the eight VS/UWS patients in the testing dataset who were initially diagnosed as VS/UWS but subsequently achieved a promising outcome.

By analyzing the sMC F-value for each predictor in the regression model, we investigated the prognostic effects of these predictors. In particular, the sMC F-value of the incidence age was greater than that of the other clinical characteristics, suggesting that incidence age might be the most important predictor among the clinical characteristics. Notably, the sMC F-value for the imaging features as a whole seemed to be greater than that for the clinical features, as shown in *Figure 4B*. We therefore speculate that the patient's residual consciousness capacity, indicated by brain networks and functional connectivity detected from resting state fMRI, might have a stronger prognostic effect than their clinical characteristics.

Some previous studies have shown that the resting state functional connectivity within the default mode network is decreased in severely brain-damaged patients, in proportion to their degree of consciousness impairment, from locked-in syndrome to minimally conscious, vegetative and coma patients (*Vanhaudenhuyse et al., 2010*). Moreover, the reduced functional connectivity within the default mode network, specifically between the MPFC and the PCC, may predict the outcome for DOC patients (*Silva et al., 2015*). Our model also validates that the aMPFC and PCC in the default mode network play important roles in predicting prognosis.

Above all, we found that the functional connectivity between the aMPFC and the DMPFC had the maximum sMC F-value in the prognostic regression model. The aMPFC is one of the core brain areas in the default mode network, whereas the DMPFC is located in the executive control network. Previous studies have demonstrated that this functional connectivity is negative connectivity in normal healthy populations, with the anti-correlation being proposed as one reflection of the intrinsic functional organization of the human brain (*Fox et al., 2005*). The default mode network directly contributes to internally generated stimulus-independent thoughts, self-monitoring, and baseline monitoring of the external world, while the executive control network supports active exploration of the external world. Correct communication and coordination between these two intrinsic anti-correlated networks is thought to be very important for optimal information integration and cognitive functioning (*Buckner et al., 2008*). A recent study reported that negative functional connectivities between the default mode network and the task-positive network were only observed in patients who recovered consciousness and in healthy controls, whereas positive values were obtained in patients with impaired consciousness (*Di Perri et al., 2016*). Further, our regression model suggests that the anti-correlations between these two diametrically opposed networks (i.e. the default mode network and the executive control network) should be the most crucial imaging feature for predicting the outcomes of the DOC patients. We therefore conclude that our prognostic model has good interpretability, and that it not only verifies the findings of previous studies but also provides a window to assess the relative significance of various predictors for the prognosis of DOC patients.

This study involved two testing datasets, which were found to be quite different, as shown in *Table 1*. First, the distributions of the etiology of the patients were remarkably different in the two datasets. The numbers of patients suffering from trauma/stroke/anoxia were 12/6/7 and 8/0/16 in the 'Beijing HDxt' and 'Guangzhou HDxt' datasets, respectively. The outcomes were also different. In the 'Beijing HDxt' dataset, 12 out of the total 25 patients recovered consciousness, compared with six out of the total 24 patients in the 'Guangzhou HDxt' dataset. Given that the characteristics of the two medical centers and their roles in the local health care system are different, we speculate that this could be one of the main reasons why the enrolled patient populations were heterogeneous. As described in the Introduction, DOC can have many causes and can be associated with several neuropathological processes and different severities, leading to the suggestion that information from different domains should be integrated to improve diagnosis and prognostication (*Bernat, 2016*). Our study demonstrates that the combination of imaging and clinical features can achieve a better performance than the use of single domain features.

However, some caution is warranted. First, although this study involved 112 DOC patients, the number of patients who would later recover consciousness was relatively low (i.e. 31/112). So, a much larger cohort will be needed for further validation. Second, the PPVs for the two testing datasets were remarkably different, with that for the 'Guangzhou HDxt' dataset being relatively low (67% versus 91%). Although our method predicted that nine patients in this dataset would recover consciousness, only six of them did (i.e. GOS $\geq$3), with the other three remaining unconscious at the end of the follow-up (i.e. GOS $\leq$2). Given that many additional factors are associated with the outcome of DOC patients, including medical complications, nutrition and so on, future work will need to integrate more information in order to provide more precise predictions. Third, the signal-to-noise ratio of resting state fMRI can influence functional connectivity analysis, so calibrations will be necessary when applying the predictive model across different sites, including standardizing the MRI acquisition protocols (e.g. scanner hardware, imaging protocols and acquisition sequences) and the patients' management strategies (see Appendix 10 for more information). Finally, a DOC patient's prognosis can be considered according to at least three dimensions: survival/mortality, recovery of consciousness, and functional recovery. This study focused on predicting recovery of consciousness, and the patients who died during the follow-up were excluded. In the future, we will consider more outcome assessments, including survival/mortality and functional recovery.

In summary, our prognostic model, which combines resting state fMRI with clinical characteristics, is proposed to predict the one-year outcome of DOC patients at an individual level. The average accuracy of classifying a patient as 'consciousness recovery' or not was around 88% in the training dataset and two unseen testing datasets. Our model also has good interpretability, thereby providing a window to reassure physicians and scientists about the significance of different predictors, including brain networks, functional connectivities and clinical characteristics. Together, these advantages could offer an objective prognosis for DOC patients that will optimize their management and deepen our understanding of brain function during unconsciousness.

## Acknowledgements

The authors appreciate the help of Ms Rowan Tweedale with the use of English in this paper. This work was partially supported by the Natural Science Foundation of China (Grant Nos. 81471380, 91432302, 31620103905), the Science Frontier Program of the Chinese Academy of Sciences (Grant No. QYZDJ-SSW-SMC019), the National Key R and D Program of China (Grant No. 2017YFA0105203), the Beijing Municipal Science and Technology Commission (Grant Nos. Z161100000216139, Z161100000216152, Z161100000516165), the Guangdong Pearl River Talents Plan Innovative and Entrepreneurial Team (grant 2016ZT06S220) and Youth Innovation Promotion Association CAS.

## Additional information

### Funding

| Funder | Grant reference number | Author |
|---|---|---|
| National Natural Science Foundation of China | 81471380 | Ming Song |
| Beijing Municipal Science and Technology Commission | Z161100000216152 | Ming Song |
| Youth Innovation Promotion Association of the Chinese Academy of Sciences | | Ming Song |
| Beijing Municipal Scienceand Technology Commission | Z161100000516165 | Yi Yang |
| National Key R&D Program of China | 2017YFA0105203 | Shan Yu Tianzi Jiang |
| National Natural Science Foundation of China | 31620103905 | Tianzi Jiang |
| National Natural Science Foundation of China | 91432302 | Tianzi Jiang |
| Chinese Academy of Sciences | Science Frontier Program: QYZDJ-SSW-SMC019 | Tianzi Jiang |
| Beijing Municipal Science and Technology Commission | Z161100000216139 | Tianzi Jiang |
| Guangdong Pearl River Talents Plan Innovative and Entrepreneurial Team | 2016ZT06S220 | Tianzi Jiang |

The funders had no role in study design, data collection and interpretation, or the decision to submit the work for publication.

### Author contributions

Ming Song, Conceptualization, Software, Formal analysis, Validation, Visualization, Methodology, Writing—original draft, Writing—review and editing; Yi Yang, Conceptualization, Data curation, Validation, Investigation, Writing—original draft, Writing—review and editing; Jianghong He, Resources, Data curation, Investigation, Writing—original draft, Writing—review and editing; Zhengyi Yang,

Formal analysis, Methodology, Writing—original draft, Writing—review and editing; Shan Yu, Methodology, Writing—original draft, Writing—review and editing; Qiuyou Xie, Data curation, Validation; Xiaoyu Xia, Yuanyuan Dang, Qiang Zhang, Xinhuai Wu, Data curation; Yue Cui, Methodology, Writing—original draft; Bing Hou, Writing—original draft, Writing—review and editing; Ronghao Yu, Ruxiang Xu, Project administration; Tianzi Jiang, Funding acquisition, Writing—original draft, Project administration, Writing—review and editing

### Author ORCIDs

Ming Song http://orcid.org/0000-0001-8846-5707
Yi Yang https://orcid.org/0000-0003-3096-0312
Shan Yu http://orcid.org/0000-0002-4385-6306
Tianzi Jiang http://orcid.org/0000-0001-9531-291X

### Ethics

Human subjects: The study was approved by the ethics committee of the PLA Army General Hospital (protocol no: 2011-097) and the ethics committee of the Guangzhou General Hospital of Guangzhou Military Command (protocol no: jz20091287). Informed consent to participate in the study was obtained from the legal surrogates of the patients and from the normal controls.

### Decision letter and Author response

Decision letter https://doi.org/10.7554/eLife.36173.058
Author response https://doi.org/10.7554/eLife.36173.059

## Additional files

### Supplementary files

• Supplementary file 1. Some examples of the warped ROIs in the default mode network for one healthy control and three DOC patients with a GOS score of 2,3 and 4, respectively.
DOI: https://doi.org/10.7554/eLife.36173.011

• Supplementary file 2. Details about single-domain prognostic models and comparisons of the single-domain and combination models.
DOI: https://doi.org/10.7554/eLife.36173.012

• Transparent reporting form
DOI: https://doi.org/10.7554/eLife.36173.013

### Data availability

We have provided anonymised demographic and clinical characteristics of the DOC patients in Appendix 1. We have made the analysis pipeline, including fMRI preprocessing, feature calculation and extraction, regression and classification, and the results visualization publicly available. Also, we have uploaded the fMRI signals in each of region of interest for every DOC patient and healthy control involved in this study. Anyone is welcome to download them from GitHub (https://github.com/realmsong504/pDOC; copy archived at https://github.com/elifesciences-publications/pDOC).

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

## Appendix 1

DOI: https://doi.org/10.7554/eLife.36173.014

### Demographic and clinical characteristics of patients and normal controls in this study

The diagnosis in this study was made by experienced physicians according to the CRS-R scale. Patients were diagnosed as MCS when they demonstrated at least one of the following behaviors: (1) following simple commands; (2) yes/no responses (gestural or verbal); (3) intelligible verbalization; (4) purposeful behavior in response to an environmental stimulus; (5) vocalization or gestures in direct response to questions; (6) reaching for objects that demonstrates a clear relationship between the position of the object and the direction of the movement; (7) touching or holding objects; (8) following or staring at an object in direct response to its movement. Emergence from the MCS was signaled by the return of functional communication and/or object use.

In this study, the patients underwent the CRS-R assessments twice weekly (or more) within the two weeks before MRI scanning. So, the CRS-R was generally administered about 4–5 times for each patient. The highest CRS-R score was considered as the diagnosis and listed in the following tables. $T_0$: the time point of the MRI scanning; $T_1$: the time point of follow-up.

**Appendix 1—table 1.** Demographic and clinical characteristics of patients in the 'Beijing_750' dataset.

| Patient alias | Gender | Age (years) | Diagnose | Etiology | Structural lesions on MRI | Time to MRI (months) | Number of CRS-R assessments | CRS-R score at $T_0$ | CRS-R subscore at $T_0$ | CRS-R score at $T_1$ | CRS-R subscore at $T_1$ | Follow-up (months) | GOS | Predicted GOS score |
|---|---|---|---|---|---|---|---|---|---|---|---|---|---|---|
| 001 | M | 36 | VS/UWS | Anoxia | Diffuse pons damage | 1 | 6 | 7 | 022102 | 22 | 446323 | 15 | 4 | 18.26 |
| 002 | M | 29 | MCS | Trauma | Bilateral-temporo-parietal damage | 9 | 4 | 18 | 355113 | 22 | 456223 | 39 | 4 | 15.31 |
| 003 | F | 33 | VS/UWS | Trauma | Bilateral-frontal lobe damage, atrophy | 12 | 5 | 7 | 102202 | 22 | 455323 | 12 | 4 | 22.88 |
| 004 | F | 28 | MCS | Trauma | L-frontal-temporal lobe damage | 1 | 4 | 15 | 335103 | 22 | 456223 | 19 | 4 | 16.58 |
| 005 | M | 23 | MCS | Anoxia | Diffuse cortical and subcortical atrophy | 3 | 4 | 10 | 232102 | 21 | 455223 | 13 | 4 | 17.08 |
| 006 | M | 45 | MCS | Stroke | L-temporo-parietal damage | 9 | 4 | 9 | 222102 | 17 | 334223 | 12 | 3 | 13.94 |
| 007 | M | 39 | MCS | Stroke | Brainstem damage | 1 | 4 | 17 | 345113 | 19 | 445123 | 12 | 3 | 14.39 |
| 008 | F | 27 | MCS | Trauma | L-basal ganglia damage | 10 | 6 | 12 | 332103 | 18 | 345123 | 19 | 3 | 16.09 |
| 009 | M | 23 | MCS | Trauma | Diffuse cortical and subcortical atrophy | 6 | 4 | 9 | 132102 | 19 | 444223 | 13 | 3 | 10.94 |
| 010 | M | 42 | MCS | Stroke | L-basal ganglia damage | 3 | 7 | 7 | 103102 | 19 | 416323 | 12 | 3 | 14.72 |
| 011 | M | 53 | MCS | Stroke | Diffuse cortical and basal ganglia (caudates) damage | 7 | 5 | 11 | 332102 | 14 | 332123 | 14 | 3 | 11.55 |
| 012 | F | 40 | VS/UWS | Stroke | Diffuse cortical and basal ganglia damage | 5 | 6 | 7 | 112102 | 14 | 333122 | 12 | 3 | 14.67 |
| 013 | M | 22 | VS/UWS | Trauma | L-frontal-temporo-parietal lobe damage | 3 | 4 | 7 | 112102 | 15 | 334122 | 27 | 3 | 15.48 |
| 014 | F | 64 | VS/UWS | Stroke | L-thalamus, basal ganglia lesions | 1 | 4 | 7 | 112102 | 11 | 233102 | 17 | 2 | 8.28 |
| 015 | F | 42 | VS/UWS | Anoxia | Diffuse anoxic cortical lesions | 1 | 4 | 7 | 112102 | 9 | 222102 | 14 | 2 | 9.02 |
| 016 | M | 45 | VS/UWS | Anoxia | Diffuse anoxic cortical lesions | 9 | 5 | 5 | 002102 | 7 | 112102 | 15 | 2 | 8.65 |

*Appendix 1—table 1 continued on next page*

Appendix 1—table 1 continued

| Patient alias | Gender | Age (years) | Diagnose | Etiology | Structural lesions on MRI | Time to MRI (months) | Number of CRS-R assessments | CRS-R score at $T_0$ | CRS-R subscore at $T_0$ | CRS-R score at $T_1$ | CRS-R subscore at $T_1$ | Follow-up (months) | GOS score | Predicted score |
|---|---|---|---|---|---|---|---|---|---|---|---|---|---|---|
| 017 | F | 60 | VS/UWS | Anoxia | Diffuse anoxic cortical lesions | 4 | 4 | 6 | 102102 | 6 | 102102 | 13 | 2 | 7.71 |
| 018 | M | 42 | VS/UWS | Stroke | R-cerebral hemisphere lesions | 6 | 4 | 7 | 112102 | 7 | 112102 | 14 | 2 | 12.44 |
| 019 | M | 51 | VS/UWS | Anoxia | Diffuse cortical and subcortical atrophy | 3 | 4 | 7 | 112102 | 7 | 112102 | 28 | 2 | 4.28 |
| 020 | F | 35 | VS/UWS | Anoxia | Bilateral-frontal lobe damage, atrophy | 2 | 4 | 7 | 112102 | 7 | 112102 | 13 | 2 | 5.87 |
| 021 | M | 71 | VS/UWS | Trauma | Diffuse cortical and subcortical atrophy | 6 | 6 | 3 | 101100 | 4 | 101101 | 13 | 2 | 4.46 |
| 022 | F | 30 | VS/UWS | Anoxia | Bilateral-basal ganglia damage | 2 | 4 | 4 | 002002 | 7 | 022102 | 38 | 2 | 6.92 |
| 023 | F | 58 | VS/UWS | Trauma | Diffuse cortical and subcortical atrophy | 2 | 4 | 3 | 002100 | 4 | 002101 | 14 | 2 | 5.09 |
| 024 | M | 23 | MCS | Trauma | R-basal ganglia (caudates) damage | 5 | 5 | 7 | 103102 | 11 | 223202 | 12 | 2 | 14.57 |
| 025 | F | 66 | VS/UWS | Trauma | Bilateral-temporo-parietal damage | 1 | 4 | 6 | 102102 | 8 | 113102 | 32 | 2 | 5.71 |
| 026 | F | 25 | VS/UWS | Anoxia | Diffuse cortical and subcortical atrophy | 3 | 4 | 5 | 102002 | 6 | 112002 | 36 | 2 | 6.75 |
| 027 | M | 48 | VS/UWS | Anoxia | Diffuse cortical and subcortical atrophy | 4 | 5 | 7 | 112102 | 8 | 113102 | 29 | 2 | 7.83 |
| 028 | F | 28 | MCS | Anoxia | Diffuse cortical and subcortical atrophy | 5 | 4 | 9 | 222102 | 11 | 233102 | 32 | 2 | 11.36 |
| 029 | M | 57 | VS/UWS | Anoxia | Diffuse cortical and subcortical atrophy | 11 | 4 | 6 | 102102 | 6 | 102102 | 33 | 2 | 4.70 |
| 030 | M | 61 | MCS | Stroke | Bilateral-temporo-parietal lobe damage | 2 | 4 | 11 | 134102 | 11 | 223112 | 12 | 2 | 10.34 |

Appendix 1—table 1 continued on next page

Appendix 1—table 1 continued

| Patient alias | Gender | Age (years) | Diagnose | Etiology | Structural lesions on MRI | Time to MRI (months) | Number of CRS-R assessments | CRS-R score at $T_0$ | CRS-R subscore at $T_0$ | CRS-R score at $T_1$ | CRS-R subscore at $T_1$ | Follow-up (months) | GOS score | Predicted score |
|---|---|---|---|---|---|---|---|---|---|---|---|---|---|---|
| 031 | M | 40 | VS/UWS | Anoxia | Diffuse cortical and subcortical atrophy | 4 | 4 | 4 | 001102 | 5 | 011102 | 27 | 2 | 5.70 |
| 032 | M | 39 | VS/UWS | Stroke | R-basal ganglia damage, atrophy | 3 | 4 | 7 | 112102 | 7 | 112102 | 12 | 2 | 8.03 |
| 033 | M | 41 | VS/UWS | Anoxia | Diffuse cortical and subcortical atrophy | 2 | 4 | 5 | 002102 | 5 | 002102 | 13 | 2 | 6.44 |
| 034 | M | 26 | VS/UWS | Stroke | Diffuse cortical and subcortical atrophy | 54 | 4 | 7 | 112102 | 7 | 112102 | 38 | 2 | 7.28 |
| 035 | F | 50 | VS/UWS | Anoxia | Diffuse cortical and subcortical atrophy | 8 | 6 | 6 | 102102 | 9 | 122202 | 12 | 2 | 5.77 |
| 036 | F | 53 | VS/UWS | Stroke | Bilateral brainstem, midbrain damage | 3 | 4 | 5 | 112100 | 7 | 112102 | 28 | 2 | 8.02 |
| 037 | M | 67 | VS/UWS | Stroke | R- brainstem, cerebellar damage | 1 | 4 | 5 | 112100 | 3 | 002001 | 12 | 2 | 2.04 |
| 038 | M | 45 | MCS | Stroke | Diffuse cortical and subcortical atrophy | 2 | 5 | 9 | 132102 | 10 | 222112 | 13 | 2 | 10.91 |
| 039 | F | 35 | VS/UWS | Anoxia | Diffuse cortical and subcortical atrophy | 3 | 4 | 6 | 102102 | 8 | 112202 | 19 | 2 | 10.24 |
| 040 | F | 46 | MCS | Trauma | Diffuse axonal injury | 77 | 7 | 11 | 222212 | 13 | 332212 | 51 | 2 | 14.76 |
| 041 | M | 49 | VS/UWS | Stroke | Bilateral-brainstem, cerebellar damage | 10 | 4 | 7 | 112102 | 7 | 112102 | 28 | 2 | 10.87 |
| 042 | M | 45 | VS/UWS | Stroke | Diffuse cortical and basal ganglia damage | 3 | 4 | 7 | 112102 | 8 | 122102 | 19 | 2 | 7.59 |
| 043 | M | 18 | VS/UWS | Anoxia | Diffuse cortical and subcortical atrophy | 8 | 5 | 6 | 111102 | 9 | 123102 | 12 | 2 | 10.85 |
| 044 | M | 53 | VS/UWS | Anoxia | Bilateral-occipital lobe damage, atrophy | 2 | 4 | 3 | 002001 | 7 | 112102 | 34 | 2 | 1.98 |
| 045 | M | 46 | VS/UWS | Trauma | R-temporo-parietal damage | 4 | 4 | 6 | 101202 | 6 | 101202 | 13 | 2 | 7.23 |

*Appendix 1—table 1 continued*

| Patient alias | Gender | Age (years) | Diagnose | Etiology | Structural lesions on MRI | Time to MRI (months) | Number of CRS-R assessments | CRS-R score at $T_0$ | CRS-R subscore at $T_0$ | CRS-R score at $T_1$ | CRS-R subscore at $T_1$ | Follow-up (months) | GOS score | Predicted score |
|---|---|---|---|---|---|---|---|---|---|---|---|---|---|---|
| 046 | F | 29 | VS/UWS | Anoxia | Diffuse cortical and subcortical atrophy | 28 | 4 | 7 | 112102 | 9 | 123102 | 12 | 2 | 8.31 |
| 047 | F | 47 | MCS | Stroke | R-basal ganglia damage | 47 | 5 | 8 | 113102 | 11 | 222212 | 12 | 2 | 9.66 |
| 048 | M | 58 | VS/UWS | Stroke | Bilateral-temporo-parietal lobe damage | 6 | 4 | 7 | 112102 | 8 | 113102 | 27 | 2 | 7.05 |
| 049 | M | 66 | VS/UWS | Anoxia | L-frontal lobe damage | 4 | 4 | 4 | 002002 | 6 | 102102 | 38 | 2 | 4.79 |
| 050 | M | 34 | VS/UWS | Trauma | Diffuse axonal injury | 3 | 4 | 6 | 112101 | 8 | 122102 | 14 | 2 | 10.28 |
| 051 | F | 31 | MCS | Trauma | L-frontal-temporo-parietal lobe damage | 3 | 5 | 11 | 133202 | 8 | 112202 | 15 | 2 | 15.56 |
| 052 | M | 33 | VS/UWS | Stroke | L-temporo-parietal lobe damage | 17 | 4 | 6 | 102102 | 8 | 113102 | 13 | 2 | 7.67 |
| 053 | F | 31 | VS/UWS | Anoxia | Diffuse cortical and basal ganglia (caudates) damage | 1 | 4 | 6 | 102102 | 6 | 102102 | 27 | 2 | 8.36 |
| 054 | F | 28 | VS/UWS | Anoxia | Diffuse cortical and subcortical atrophy | 3 | 4 | 6 | 102102 | 8 | 112202 | 32 | 2 | 9.23 |
| 055 | F | 26 | VS/UWS | Stroke | L-basal ganglia damage | 4 | 4 | 6 | 102102 | 6 | 102102 | 12 | 2 | 10.96 |
| 056 | M | 45 | VS/UWS | Trauma | Diffuse axonal injury | 1 | 4 | 6 | 102102 | 6 | 102102 | 29 | 2 | 9.05 |
| 057 | F | 69 | VS/UWS | Stroke | Diffuse cortical and subcortical atrophy | 4 | 4 | 6 | 102102 | 7 | 102202 | 33 | 2 | 12.43 |
| 058 | F | 68 | VS/UWS | Trauma | Diffuse axonal injury | 6 | 6 | 7 | 112102 | 9 | 132102 | 17 | 2 | 9.74 |
| 059 | M | 50 | VS/UWS | Stroke | L-frontal-temporo-parietal lobe damage | 3 | 4 | 6 | 111102 | 8 | 222002 | 27 | 2 | 7.01 |
| 060 | M | 60 | MCS | Trauma | Bilateral brainstem, midbrain damage | 7 | 4 | 11 | 134102 | 11 | 223112 | 30 | 2 | 11.69 |
| 061 | M | 44 | VS/UWS | Anoxia | Diffuse cortical and subcortical atrophy | 2 | 4 | 6 | 102102 | 4 | 002101 | 13 | 2 | 10.48 |

*Appendix 1—table 1 continued on next page*

*Appendix 1—table 1 continued*

| Patient alias | Gender | Age (years) | Diagnose | Etiology | Structural lesions on MRI | Time to MRI (months) | Number of CRS-R assessments | CRS-R score at $T_0$ | CRS-R subscore at $T_0$ | CRS-R score at $T_1$ | CRS-R subscore at $T_1$ | Follow-up (months) | GOS | Predicted score |
|---|---|---|---|---|---|---|---|---|---|---|---|---|---|---|
| 062 | F | 35 | VS/UWS | Anoxia | Bilateral-basal ganglia damage | 3 | 5 | 7 | 211102 | 9 | 231102 | 27 | 2 | 9.07 |
| 063 | F | 43 | VS/UWS | Anoxia | Diffuse cortical and subcortical atrophy | 2 | 4 | 7 | 112102 | 8 | 202112 | 29 | 2 | 10.09 |

DOI: https://doi.org/10.7554/eLife.36173.015

**Appendix 1—table 2.** Demographic and clinical characteristics of patients in the 'Beijing_HDxt' dataset.

| Patient alias | Gender | Age (years) | Diagnose | Etiology | Structural lesions on MRI | Time to MRI (months) | Number of CRS-R assessments | CRS-R score at T0 | CRS-R subscore at T0 | CRS-R score at T1 | CRS-R subscore at T1 | follow-up (months) | GOS | Predicted GOS score |
|---|---|---|---|---|---|---|---|---|---|---|---|---|---|---|
| 001 | M | 19 | VS/UWS | Trauma | L-temporo-parietal lobe damage | 6 | 4 | 7 | 112102 | 22 | 456223 | 40 | 4 | 20.37 |
| 002 | M | 26 | MCS | Trauma | R-thalamus, basal ganglia lesions | 3 | 6 | 10 | 232102 | 23 | 456323 | 47 | 4 | 17.12 |
| 003 | F | 22 | VS/UWS | Trauma | L-temporal lobe damage | 4 | 4 | 6 | 102102 | 22 | 456223 | 47 | 4 | 14.05 |
| 004 | M | 41 | VS/UWS | Stroke | Bilateral brainstem, midbrain damage | 3 | 4 | 6 | 112101 | 23 | 456323 | 50 | 4 | 20.23 |
| 005 | M | 36 | MCS | Stroke | Bilateral brainstem damage | 4 | 4 | 6 | 003102 | 23 | 456323 | 39 | 4 | 7.75 |
| 006 | M | 34 | VS/UWS | Anoxia | Diffuse cortical and subcortical atrophy | 1 | 4 | 6 | 111102 | 14 | 323123 | 31 | 3 | 17.25 |
| 007 | F | 18 | VS/UWS | Trauma | Diffuse axonal injury | 3 | 4 | 5 | 012002 | 14 | 332123 | 41 | 3 | 14.86 |
| 008 | M | 58 | MCS | Trauma | R-frontal lobe damage | 12 | 4 | 8 | 113102 | 15 | 333123 | 40 | 3 | 15.62 |
| 009 | M | 41 | MCS | Trauma | R-frontal-temporo-parietal lobe damage | 1 | 5 | 11 | 233012 | 18 | 344223 | 42 | 3 | 18.89 |
| 010 | M | 46 | VS/UWS | Stroke | L-brainstem, cerebellar damage | 7 | 4 | 6 | 102102 | 14 | 332123 | 53 | 3 | 17.05 |
| 011 | M | 25 | VS/UWS | Anoxia | Diffuse cortical and subcortical atrophy | 4 | 6 | 6 | 102102 | 14 | 224123 | 46 | 3 | 18.07 |
| 012 | M | 58 | VS/UWS | Trauma | L-brainstem damage | 1 | 7 | 7 | 112102 | 19 | 355123 | 40 | 3 | 10.75 |
| 013 | M | 36 | VS/UWS | Trauma | L-frontal-temporo-parietal lobe damage | 6 | 4 | 7 | 112102 | 10 | 232102 | 44 | 2 | 9.58 |
| 014 | M | 58 | VS/UWS | Trauma | R-frontal-temporo-parietal lobe damage | 4 | 4 | 6 | 102102 | 6 | 102102 | 45 | 2 | 6.69 |
| 015 | M | 65 | VS/UWS | Stroke | Diffuse cortical and subcortical atrophy | 3 | 4 | 3 | 100002 | 5 | 101102 | 43 | 2 | 4.01 |
| 016 | F | 24 | VS/UWS | Trauma | Diffuse axonal injury | 44 | 6 | 6 | 102102 | 8 | 122102 | 44 | 2 | 14.03 |

*Appendix 1—table 2 continued on next page*

Appendix 1—table 2 continued

| Patient alias | Gender | Age (years) | Diagnose | Etiology | Structural lesions on MRI | Time to MRI (months) | Number of CRS-R assessments | CRS-R score at T0 | CRS-R subscore at T0 | CRS-R score at T1 | CRS-R subscore at T1 | follow-up (months) | GOS | Predicted GOS score |
|---|---|---|---|---|---|---|---|---|---|---|---|---|---|---|
| 017 | F | 46 | VS/UWS | Stroke | L-pons damage | 2 | 4 | 7 | 112102 | 7 | 112102 | 40 | 2 | 12.11 |
| 018 | M | 53 | VS/UWS | Anoxia | Diffuse cortical and subcortical atrophy | 3 | 4 | 4 | 101002 | 6 | 102102 | 41 | 2 | 5.38 |
| 019 | F | 32 | VS/UWS | Trauma | L-temporo-parietal lobe damage | 3 | 4 | 6 | 102102 | 8 | 112202 | 23 | 2 | 13.76 |
| 020 | M | 41 | VS/UWS | Anoxia | Diffuse cortical and subcortical atrophy | 2 | 4 | 4 | 101002 | 8 | 112202 | 40 | 2 | 12.06 |
| 021 | F | 33 | VS/UWS | Anoxia | Diffuse cortical and subcortical atrophy | 7 | 5 | 6 | 211002 | 11 | 232202 | 47 | 2 | 4.55 |
| 022 | M | 49 | VS/UWS | Anoxia | Diffuse cortical and subcortical atrophy | 2 | 4 | 6 | 102102 | 6 | 102102 | 14 | 2 | 8.97 |
| 023 | F | 25 | MCS | Anoxia | Bilateral thalamus, brainstem damage | 4 | 7 | 14 | 450023 | 10 | 240022 | 50 | 2 | 12.42 |
| 024 | M | 63 | VS/UWS | Stroke | L-basal ganglia lesions | 5 | 4 | 4 | 001102 | 5 | 101102 | 48 | 2 | 8.22 |
| 025 | M | 68 | VS/UWS | Trauma | L-frontal-temporo-parietal lobe damage | 2 | 4 | 5 | 002102 | 6 | 012102 | 47 | 2 | 9.72 |

DOI: https://doi.org/10.7554/eLife.36173.016

**Appendix 1—table 3.** Demographic and clinical characteristics of patients in the 'Guangzhou_HDxt' dataset.

| Patient alias | Gender | Age (years) | Diagnose | Etiology | Time to MRI (months) | CRS-R score at $T_0$ | CRS-R subscore at $T_0$ | Follow-up (months) | GOS | Predicted score |
|---|---|---|---|---|---|---|---|---|---|---|
| 001 | F | 15 | MCS | Anoxia | 1 | 13 | 135112 | 59 | 4 | 20.92 |
| 002 | M | 29 | MCS | Trauma | 4 | 9 | 114012 | 61 | 3 | 16.50 |
| 003 | F | 27 | MCS | Trauma | 1 | 9 | 105102 | 29 | 3 | 16.67 |
| 004 | F | 20 | MCS | Trauma | 2 | 8 | 113102 | 41 | 3 | 20.37 |
| 005 | M | 30 | MCS | Trauma | 1 | 15 | 116223 | 63 | 3 | 17.34 |
| 006 | M | 31 | MCS | Trauma | 1 | 20 | 445223 | 51 | 3 | 14.00 |
| 007 | M | 28 | VS/UWS | Anoxia | 1 | 5 | 102002 | 69 | 2 | 9.71 |
| 008 | M | 48 | MCS | Trauma | 1 | 12 | 234102 | 55 | 2 | 9.11 |
| 009 | M | 46 | VS/UWS | Trauma | 2 | 4 | 102001 | 65 | 2 | 12.58 |
| 010 | M | 78 | VS/UWS | Anoxia | 1 | 4 | 100102 | 49 | 2 | 9.20 |
| 011 | M | 39 | VS/UWS | Anoxia | 1 | 5 | 002102 | 51 | 2 | 4.20 |
| 012 | F | 46 | VS/UWS | Anoxia | 2 | 4 | 001102 | 46 | 2 | 14.49 |
| 013 | M | 39 | VS/UWS | Anoxia | 2 | 5 | 102002 | 43 | 2 | 7.30 |
| 014 | M | 16 | VS/UWS | Anoxia | 2 | 3 | 001002 | 71 | 2 | 14.17 |
| 015 | M | 25 | MCS | Anoxia | 1 | 12 | 135102 | 78 | 2 | 8.56 |
| 016 | F | 76 | VS/UWS | Anoxia | 5 | 4 | 100102 | 59 | 2 | 3.57 |
| 017 | M | 36 | VS/UWS | Trauma | 2 | 5 | 001202 | 65 | 2 | 17.28 |
| 018 | M | 32 | VS/UWS | Anoxia | 10 | 6 | 102102 | 56 | 2 | 5.66 |
| 019 | F | 49 | VS/UWS | Anoxia | 1 | 6 | 102102 | 49 | 2 | 4.55 |
| 020 | F | 52 | VS/UWS | Anoxia | 1 | 3 | 000102 | 27 | 2 | 5.08 |
| 021 | M | 62 | VS/UWS | Anoxia | 2 | 3 | 001002 | 28 | 2 | 10.26 |
| 022 | F | 33 | VS/UWS | Anoxia | 2 | 6 | 102102 | 29 | 2 | 9.09 |
| 023 | F | 28 | VS/UWS | Anoxia | 9 | 6 | 102102 | 67 | 2 | 12.62 |
| 024 | F | 57 | VS/UWS | Anoxia | 1 | 5 | 002102 | 42 | 2 | 4.72 |

DOI: https://doi.org/10.7554/eLife.36173.017

**Appendix 1—table 4.** Demographic of healthy controls in the 'Beijing_750' dataset.

| Alias | Gender | Age | Handedness |
|---|---|---|---|
| NC001 | F | 40 | Right |
| NC002 | M | 50 | Right |
| NC003 | F | 34 | Right |
| NC004 | M | 25 | Right |
| NC005 | M | 28 | Right |
| NC007 | F | 24 | Right |
| NC008 | F | 47 | Right |
| NC009 | F | 22 | Right |
| NC010 | F | 60 | Right |
| NC012 | F | 26 | Right |
| NC013 | M | 21 | Right |
| NC014 | F | 27 | Right |
| NC015 | M | 40 | Right |
| NC016 | M | 44 | Right |
| NC017 | F | 22 | Right |
| NC018 | M | 50 | Right |
| NC019 | M | 27 | Right |
| NC020 | F | 43 | Right |
| NC021 | F | 25 | Right |
| NC022 | M | 54 | Right |
| NC023 | F | 52 | Right |
| NC026 | M | 46 | Right |
| NC027 | F | 52 | Right |
| NC028 | M | 29 | Right |
| NC029 | F | 46 | Right |
| NC030 | M | 44 | Right |
| NC031 | M | 30 | Right |
| NC032 | M | 31 | Right |
| NC033 | M | 32 | Right |
| NC034 | M | 30 | Right |

DOI: https://doi.org/10.7554/eLife.36173.018

**Appendix 1—table 5.** Demographic of healthy controls in the 'Beijing_HDxt' dataset.

| Alias | Gender | Age | Handedness |
|---|---|---|---|
| NC001_HDxt | M | 44 | Right |
| NC002_HDxt | M | 42 | Right |
| NC003_HDxt | M | 30 | Right |
| NC004_HDxt | M | 40 | Right |
| NC005_HDxt | M | 30 | Right |
| NC006_HDxt | M | 30 | Right |
| NC007_HDxt | F | 58 | Right |
| NC008_HDxt | F | 54 | Right |
| NC009_HDxt | F | 41 | Right |
| NC010_HDxt | F | 41 | Right |

DOI: https://doi.org/10.7554/eLife.36173.019

## Appendix 2

DOI: https://doi.org/10.7554/eLife.36173.020

### Brain networks and regions of interest in this study

The six brain networks investigated in this study and the names of regions of interest (ROI). The *Appendix 2—table 1* represented the six brain networks, the name of ROIs, the peak coordinates in the Montreal Neurological Institute (MNI) space and the corresponding references. All of ROI were defined as a spherical region with a radius of 6 mm at the center of the peak coordinates of the ROI.

**Appendix 2—table 1.** Brain networks and ROIs in this study.

| Brain network | ROI name | ROI abbreviation | Peak MNI coordinates | References |
|---|---|---|---|---|
| Default mode | | | | (*Raichle, 2011*; *Demertzi et al., 2015*) |
| | Anterior medial pre-frontal cortex | aMPFC | −1 54 27 | |
| | Posterior cingulate cortex/precuneus | PCC | 0 –52 27 | |
| | Left lateral parietal cortex | L.LatP | −46 –66 30 | |
| | Right lateral parietal cortex | R.LatP | 49 –63 33 | |
| Executive control | | | | (*Seeley et al., 2007*; *Raichle, 2011*) |
| | Dorsal medial PFC | DMPFC | 0 27 46 | |
| | Left anterior prefrontal cortex | L.PFC | −44 45 0 | |
| | Right anterior prefrontal cortex | R.PFC | 44 45 0 | |
| | Left superior parietal cortex | L. Parietal | −50 –51 45 | |
| | Right superior parietal cortex | R. Parietal | 50 –51 45 | |
| Salience | | | | (*Seeley et al., 2007*; *Raichle, 2011*; *Demertzi et al., 2015*) |
| | Left orbital frontoinsula | L.AIns | −40 18 –12 | |
| | Right orbital frontoinsula | R.AIns | 42 10 –12 | |
| | Dorsal anterior cingulate | dACC | 0 18 30 | |
| Sensorimotor | | | | (*Raichle, 2011*; *Demertzi et al., 2015*) |
| | Left primary motor cortex | L.M1 | −39 –26 51 | |
| | Right primary motor cortex | R.M1 | 38 –26 51 | |
| | Supplementary motor area | SMA | 0 –21 51 | |
| Auditory | | | | (*Raichle, 2011*; *Demertzi et al., 2015*) |

*Appendix 2—table 1 continued on next page*

*Appendix 2—table 1 continued*

| Brain network | ROI name | ROI abbreviation | Peak MNI coordinates | References |
|---|---|---|---|---|
| | Left primary auditory cortex | L.A1 | −62 −30 12 | |
| | Right primary auditory cortex | R.A1 | 59 −27 15 | |
| | Middle cingulate cortex | MCC | 0 −7 43 | |
| Visual | | | | (*Demertzi et al., 2015*) |
| | Left primary visual cortex | L.V1 | −13 −85 6 | |
| | Right primary visual cortex | R.V1 | 8 −82 6 | |
| | Left associative visual cortex | L.V4 | −30 −89 20 | |
| | Right associative visual cortex | R.V4 | 30 −89 20 | |

DOI: https://doi.org/10.7554/eLife.36173.021

## Appendix 3

DOI: https://doi.org/10.7554/eLife.36173.022

### Brain functional network templates

Although the neurobiological implications of the spontaneous neuronal activity are not very clear, spontaneous fluctuations in the blood oxygenation level-dependent signal have been found to be coherent within a variety of functionally relevant brain regions, which are denoted as representing a 'network'. Moreover, several networks have been found to be spatially consistent across different healthy subjects (*Damoiseaux et al., 2006*). Researchers suggested that the brain networks assessed by resting state fMRI may reflect an intrinsic functional architecture of the brain (*Raichle, 2011*). As mentioned in the manuscript, multiple networks were reported to be disrupted in the DOC patients. Here, the connection templates of the six brain networks were investigated within the healthy control group of the 'Beijing 750' dataset. This study focused on the cortex, so six functional networks were investigated, including the default mode network, the executive control network, salience, and the sensorimotor, auditory, and visual networks. Group functional connectivity maps for each of the six networks were created with a one-sample t test as shown in the following *Appendix 3—figure 1*. These templates were separately shown on the brain surface using the SurfStat toolbox (SurfStat, RRID:SCR_007081). The color bar represents the T value.

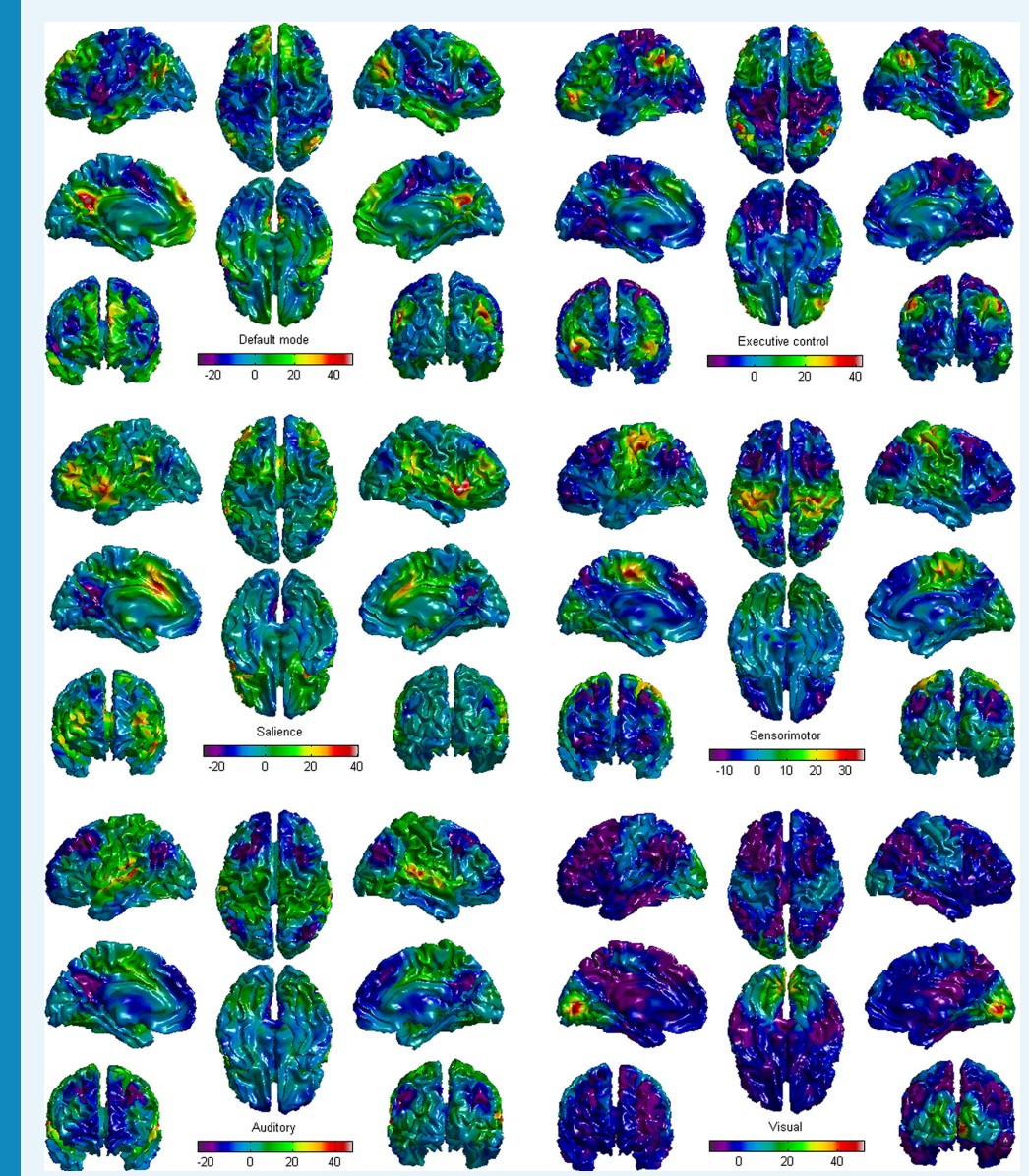

**Appendix 3—figure 1.** The six brain functional network templates in this study.
DOI: https://doi.org/10.7554/eLife.36173.023

# Appendix 4

DOI: https://doi.org/10.7554/eLife.36173.024

## Quality control for resting state functional connectivity

During the MRI scanning, the foam pad and headphones were used to reduce head motion and scanner noise. The normal controls were instructed to keep still with their eyes closed, as motionless as possible and not to think about anything in particular. The same instructions were given to the patients but due to their consciousness and cognitive impairments, we could not fully control for a prolonged eye-closed yet awake scanning session. The *Appendix 4—figure 1* showed cumulative distribution of head motion per volume (framewise displacement) for normal controls and the patients. The *Appendix 4—figure 2* showed the results of control quality of resting state fMRI in this study. The *Appendix 4—figure 3* showed the histogram of the remaining number of fMRI volumes after scrubbing.

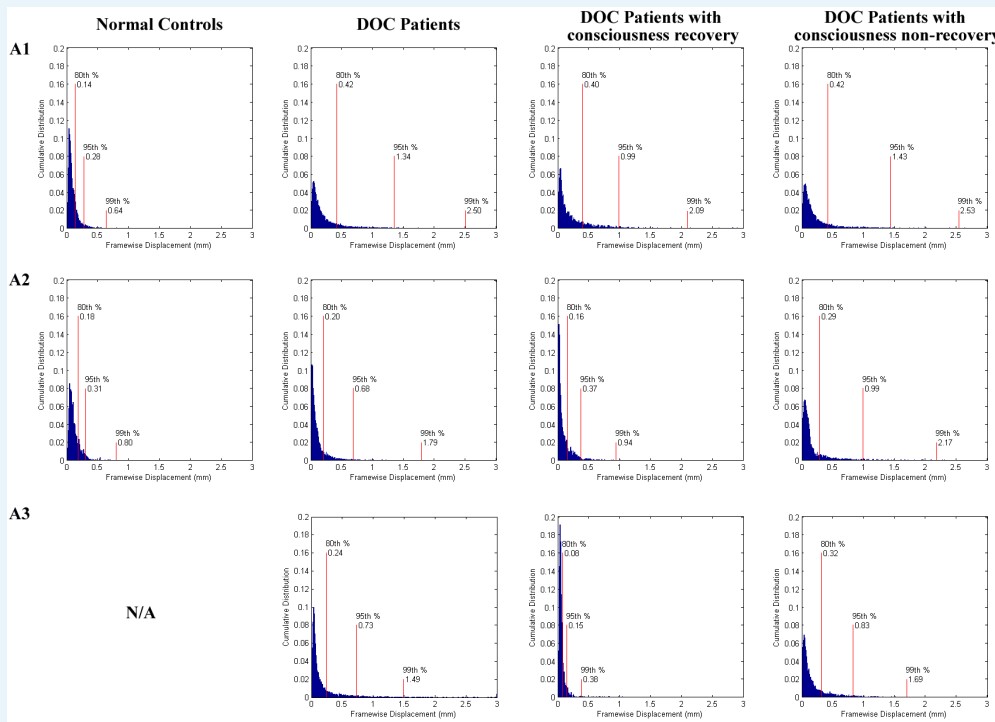

**Appendix 4—figure 1.** Cumulative distribution of head motion per volume (framewise displacement) for normal controls and DOC patients separately in the training dataset 'Beijing 750' (**A1**), the testing dataset 'Beijing HDxt' (**A2**), and the testing dataset 'Guangzhou HDxt' (**A3**). The normal controls were shown in left column, whereas the DOC patients were shown in right column. No healthy control data were available for the Guangzhou centre. In both patients and controls, head position was stable to within 1.5 mm for the vast majority (>95%) of brain volumes.

DOI: https://doi.org/10.7554/eLife.36173.025

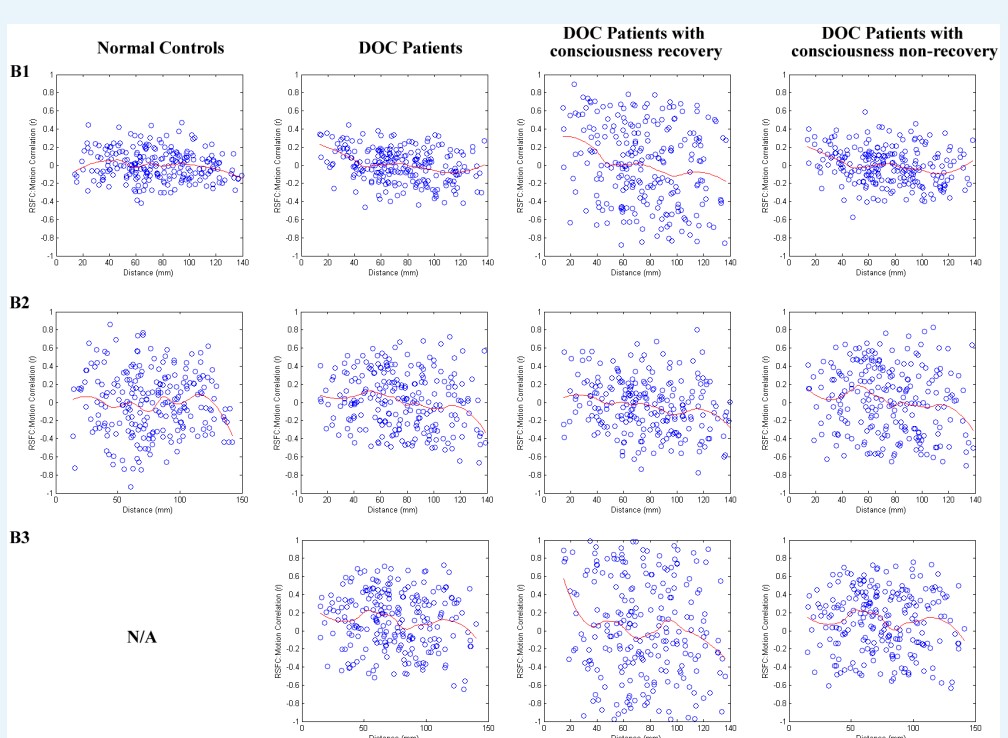

**Appendix 4—figure 2.** Correlations between motion artifact and neuroanatomical distance between the ROIs in this study. Prior studies have shown that motion artifacts tend to vary with neuroanatomical distance between brain nodes. Here, we conducted quality control analyses as described in the previous study (*Power et al., 2015*). Specifically, we computed correlations between head motion (mean FD) and each resting state functional connectivity (RSFC) feature and plotted them as a function of neuroanatomical distance (mm) for subjects in the training dataset 'Beijing 750' (**B1**), the testing dataset 'Beijing HDxt' (**B2**), and the testing dataset 'Guangzhou HDxt' (**B3**). Smoothing curves (in red) were plotted using a moving average filter.

DOI: https://doi.org/10.7554/eLife.36173.026

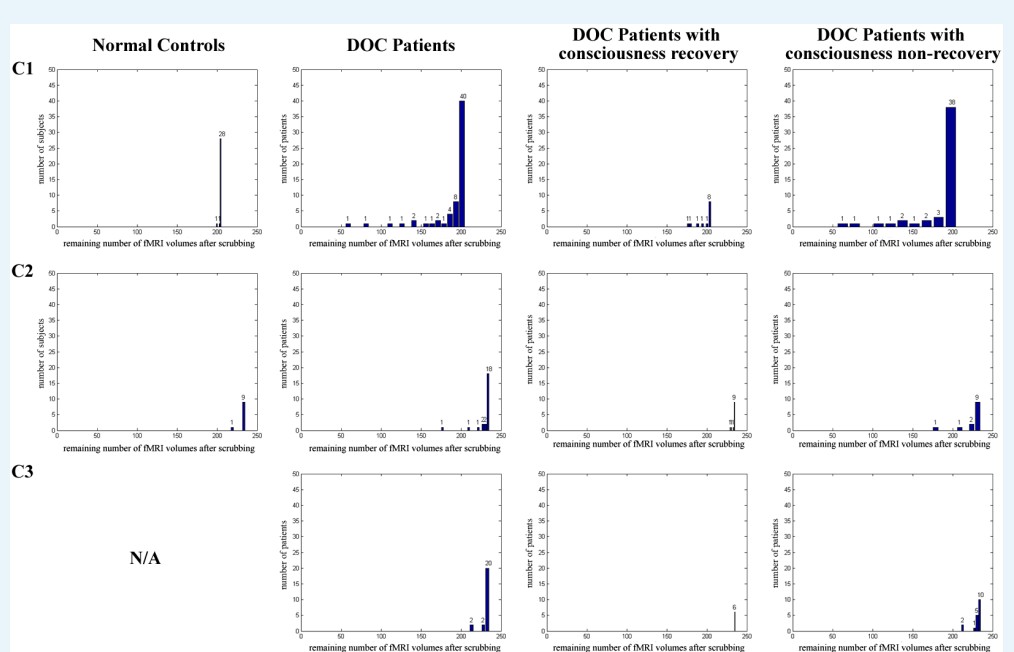

**Appendix 4—figure 3.** Histogram of the remaining number of fMRI volumes after scrubbing for each population, specifically 'Beijing 750' datatset (**C1**), 'Beijing HDxt' dataset (**C2**), and 'Guangzhou HDxt' dataset (**C3**).

DOI: https://doi.org/10.7554/eLife.36173.027

## Appendix 5

DOI: https://doi.org/10.7554/eLife.36173.028

### Warped regions of interest and brain network templates

The conventional fMRI preprocess normalizes individual fMRI images into a standard space defined by a specific template image. This study generated a functional connectivity image for each patient in his/her own fMRI space. During the preprocessing of each patient's fMRI scans, the 22 ROIs and the six brain network templates were spatially warped to individual fMRI space and resampled to the voxel size of the individual fMRI image. To ensure the registration, we developed some tools to check the transformed ROIs and brain network templates visually for each subject in this study.

*Supplementary file 1* illustrated some examples of the warped ROIs in the default mode network (DMN) for the three DOC patients with a GOS score of 2,3 or 4, respectively. In addition, as a reference, we showed these figures for one normal control. The ROIs in the DMN include the anterior medial prefrontal cortex (aMPFC), the posterior cingulate cortex/precuneus (PCC), the left lateral parietal cortex (L.LatP), the right lateral parietal cortex (R.LatP). The details about these four ROIs are listed in Appendix 2, and the brain network template of the DMN is provided in Appendix 3.

## Appendix 6

DOI: https://doi.org/10.7554/eLife.36173.029

# Correlations between imaging features and CRS-R scores at $T_1$.

In addition, *Appendix 6—figure 1* illuminated these brain area connection features and their Pearson's correlations to the CRS-R scores at the $T_1$ time point.

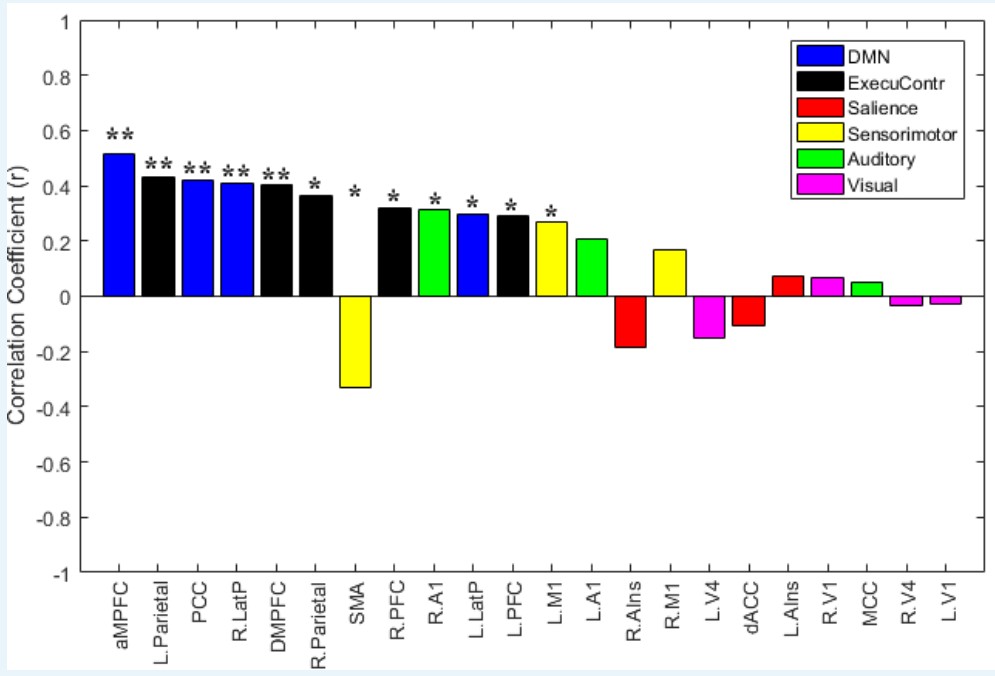

**Appendix 6—figure 1.** The brain area connection features sorted by their Pearson's correlations to the CRS-R scores at the $T_1$ time point in the training dataset 'Beijing 750'.

DOI: https://doi.org/10.7554/eLife.36173.030

In addition, *Appendix 6—figure 2* illuminates the functional connectivity features that were significantly correlated to the CRS-R scores at the $T_1$ time point.

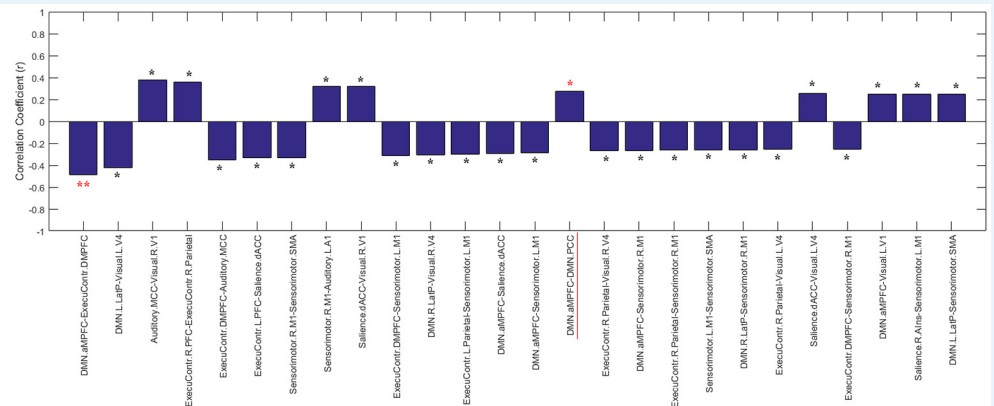

**Appendix 6—figure 2.** The functional connectivity features sorted by their Pearson's correlations to the CRS-R scores at the $T_1$ time point across the DOC patients in the training dataset

'Beijing 750'.
DOI: https://doi.org/10.7554/eLife.36173.031

   *Appendix 6—figure 3* showed these significant functional connectivity features in a Circos manner. The red links represent the within-network functional connectivity, while the blue links represent the inter-network functional connectivity. The width of a link is proportional to the strength of the functional connectivity.

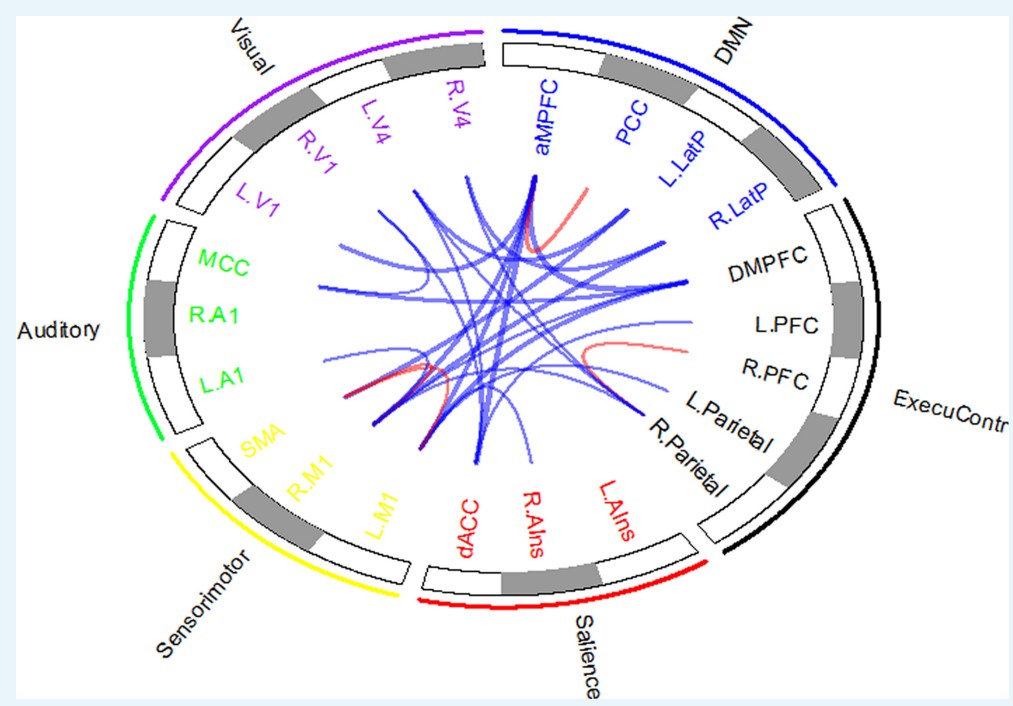

**Appendix 6—figure 3.** The Circos map for the functional connectivity features that were significantly correlated to the CRS-R scores at the $T_1$ time point across the DOC patients in the training dataset 'Beijing 750'.
DOI: https://doi.org/10.7554/eLife.36173.032

**Appendix 6—table 1.** The brain area connection features and their Pearson's correlations to the CRS-R scores at the $T_1$ time point across the DOC patients in the training dataset 'Beijing 750'.

|  | ROI name | Pearson's correlation coefficient and p value |
|---|---|---|
| ** | DMN.aMPFC | r = 0.514, p=0.000 |
| ** | ExecuContr.L.Parietal | r = 0.429, p=0.000 |
| ** | DMN.PCC | r = 0.420, p=0.001 |
| ** | DMN.R.LatP | r = 0.407, p=0.001 |
| ** | ExecuContr.DMPFC | r = 0.405, p=0.001 |
| * | ExecuContr.R.Parietal | r = 0.363, p=0.003 |
| * | Sensorimotor.SMA | r = −0.332, p=0.008 |
| * | ExecuContr.R.PFC | r = 0.320, p=0.011 |
| * | Auditory.R.A1 | r = 0.315, p=0.012 |
| * | DMN.L.LatP | r = 0.298, p=0.018 |
| * | ExecuContr.L.PFC | r = 0.291, p=0.021 |
| * | Sensorimotor.L.M1 | r = 0.267, p=0.035 |

*Appendix 6—table 1 continued on next page*

*Appendix 6—table 1 continued*

| ROI name | Pearson's correlation coefficient and p value |
|---|---|
| Auditory.L.A1 | r = 0.206, p=0.105 |
| Salience.R.AIns | r = −0.187, p=0.142 |
| Sensorimotor.R.M1 | r = 0.167, p=0.191 |
| Visual.L.V4 | r = −0.151, p=0.236 |
| Salience.dACC | r = −0.104, p=0.418 |
| Salience.L.AIns | r = 0.075, p=0.560 |
| Visual.R.V1 | r = 0.065, p=0.611 |
| Auditory.MCC | r = 0.053, p=0.682 |
| Visual.R.V4 | r = −0.031, p=0.809 |
| Visual.L.V1 | r = −0.028, p=0.830 |

**: $p < 0.05$, FDR corrected; *: $p < 0.05$, uncorrected.

DOI: https://doi.org/10.7554/eLife.36173.033

**Appendix 6—table 2.** Functional connectivity features and their Pearson's correlations to the CRS-R scores at the $T_1$ time point across the DOC patients in the training dataset 'Beijing 750'.

| | Functional connectivity | Pearson's correlation coefficient and p-value |
|---|---|---|
| † | DMN.aMPFC - ExecuContr.DMPFC | r = −0.489, p=0.000 |
| * | DMN.L.LatP - Visual.L.V4 | r = −0.421, p=0.001 |
| * | Auditory.MCC - Visual.R.V1 | r = 0.375, p=0.002 |
| * | ExecuContr.R.PFC - ExecuContr.R.Parietal | r = 0.361, p=0.004 |
| * | ExecuContr.DMPFC - Auditory.MCC | r = −0.351, p=0.005 |
| * | ExecuContr.L.PFC - Salience.dACC | r = −0.335, p=0.007 |
| * | Sensorimotor.R.M1 - Sensorimotor.SMA | r = −0.330, p=0.008 |
| * | Sensorimotor.R.M1 - Auditory.L.A1 | r = 0.319, p=0.011 |
| * | Salience.dACC - Visual.R.V1 | r = 0.319, p=0.011 |
| * | ExecuContr.DMPFC - Sensorimotor.L.M1 | r = −0.310, p=0.013 |
| * | DMN.R.LatP - Visual.R.V4 | r = −0.306, p=0.015 |
| * | ExecuContr.L.Parietal - Sensorimotor.L.M1 | r = −0.302, p=0.016 |
| * | DMN.aMPFC - Salience.dACC | r = −0.292, p=0.020 |
| * | DMN.aMPFC - Sensorimotor.L.M1 | r = −0.286, p=0.023 |
| * | DMN.aMPFC - DMN.PCC | r = 0.275, p=0.029 |
| * | ExecuContr.R.Parietal - Visual.R.V4 | r = −0.270, p=0.033 |
| * | DMN.aMPFC - Sensorimotor.R.M1 | r = −0.268, p=0.034 |
| * | ExecuContr.R.Parietal - Sensorimotor.R.M1 | r = −0.263, p=0.037 |
| * | Sensorimotor.L.M1 - Sensorimotor.SMA | r = −0.261, p=0.039 |
| * | DMN.R.LatP - Sensorimotor.R.M1 | r = −0.261, p=0.039 |
| * | ExecuContr.R.Parietal - Visual.L.V4 | r = −0.257, p=0.042 |
| * | Salience.dACC - Visual.L.V4 | r = 0.256, p=0.043 |
| ** | ExecuContr.DMPFC - Sensorimotor.R.M1 | r = −0.255, p=0.043 |
| * | DMN.aMPFC - Visual.L.V1 | r = 0.251, p=0.047 |
| * | Salience.R.AIns - Sensorimotor.L.M1 | r = 0.250, p=0.049 |
| * | DMN.L.LatP - Sensorimotor.SMA | r = 0.248, p=0.050 |

Specifically, the functional connectivity features were the functional connectivity between any pair of ROIs. As there were more than 200 functional connectivity features, because of space limitations, only the functional connectivity features that were significantly correlated to the CRS-R scores at the $T_1$ time point are shown. **: $p < 0.05$, FDR corrected; *: $p < 0.05$, uncorrected.

DOI: https://doi.org/10.7554/eLife.36173.034

## Appendix 7

DOI: https://doi.org/10.7554/eLife.36173.035

### Histogram depicting the imaging features included in CARS-PLSR models

We resampled 1000 times with replacement from the training dataset 'Beijing 750'. In each bootstrap sampling set, the CARS-PLSR was used for imaging feature subset selection. We then summarized the numbers of each imaging feature that was included in the CARS-PLSR model. *Appendix 7—figure 1* shows the histogram depicting the imaging features included in CARS-PLSR models. The horizontal bar represents the number.

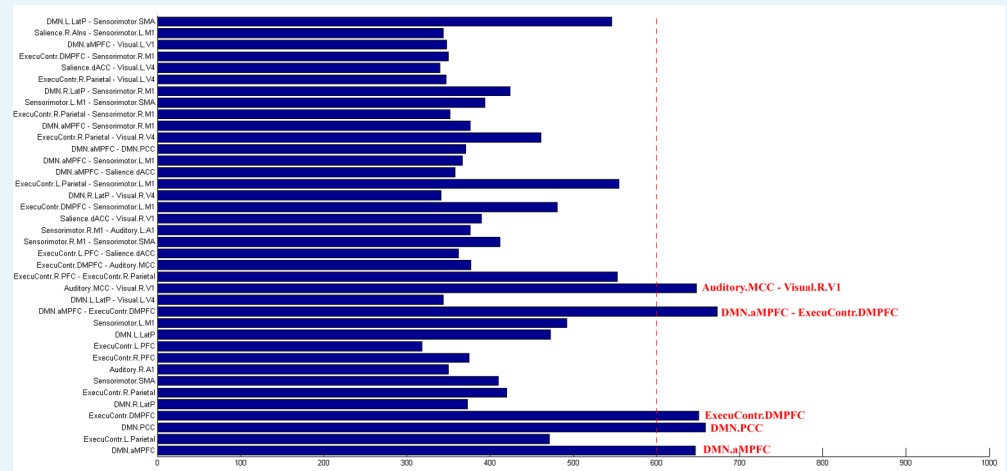

**Appendix 7—figure 1.** Histogram depicting the imaging features included in CARS-PLSR models.

DOI: https://doi.org/10.7554/eLife.36173.036

# Appendix 8

DOI: https://doi.org/10.7554/eLife.36173.037

## Validations in healthy controls

To test robustness, we evaluated whether the prognostic regression model generalized to the normal controls (NC) in the training dataset 'Beijing 750' (n = 30) and the testing dataset 'Beijing HDxt' (n = 10). No normal control data were available in the Guangzhou centre. As the NC subjects did not have the clinical characteristics, we calculated the subscores using the imaging features alone and then compared these subscores to those of the DOC patients. *Appendix 8—figure 1* showed the imaging subscores for all of the subjects in the three datasets. We would like to emphasize that the normal controls in the training dataset were only used to establish the brain network templates, and not for any training.

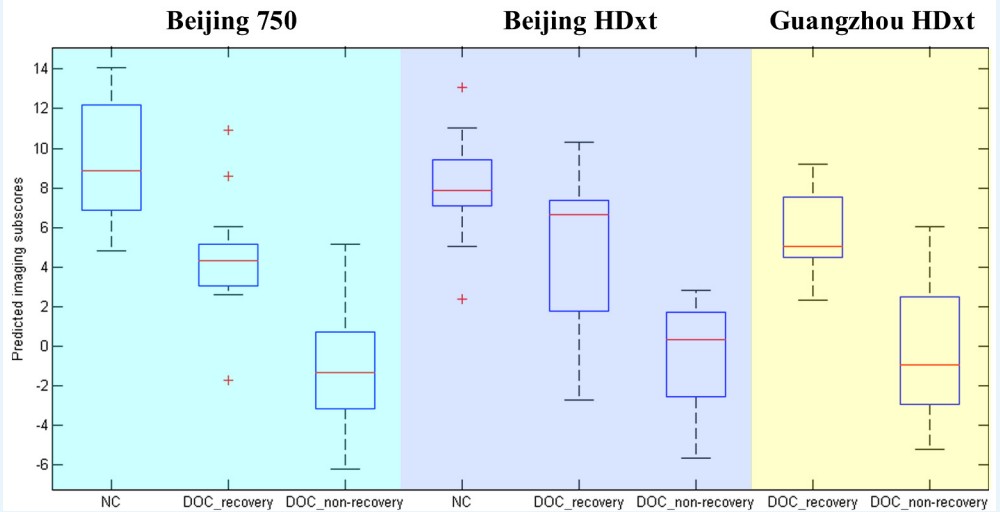

**Appendix 8—figure 1.** The imaging subscores for all of the subjects in the three datasets.
DOI: https://doi.org/10.7554/eLife.36173.038

We found that (1) in the training dataset 'Beijing 750', the NC subjects had significantly larger imaging subscores in comparison to both the DOC patients with consciousness recovery and the DOC patients with consciousness non-recovery (one-way ANOVA, p<0.05, multiple comparison corrected), and that the DOC patients with consciousness recovery had significantly larger imaging subscores in comparison to the DOC patients with consciousness non-recovery (one-way ANOVA, p<0.05, multiple comparison corrected); (2) in the testing dataset 'Beijing HDxt', the NC subjects had significantly larger imaging subscores in comparison to the DOC patients with consciousness non-recovery (one-way ANOVA, p<0.05, multiple comparison corrected), and the DOC patients with consciousness recovery had significantly larger imaging subscores in comparison to the DOC patients with consciousness non-recovery (one-way ANOVA, p<0.05, multiple comparison corrected); (3) in the testing dataset 'Guangzhou HDxt', the imaging subscores of the DOC patients with consciousness recovery were significantly larger than that of DOC patients with consciousness non-recovery (two-sample t-tests, p<0.05).

## Appendix 9

DOI: https://doi.org/10.7554/eLife.36173.039

### Variations across different sites

To investigate variations across different sites, we did two experiments using the normal control (NC) subjects in this study. First, we explored whether the predicted imaging subscores of the NC subjects were significantly different between the training dataset 'Beijing 750' (n = 30) and the testing dataset 'Beijing HDxt' (n = 10). We found that there was no significant difference between the two groups (two-sample T test, p=0.24). The distribution is shown in *Appendix 9—figure 1*.

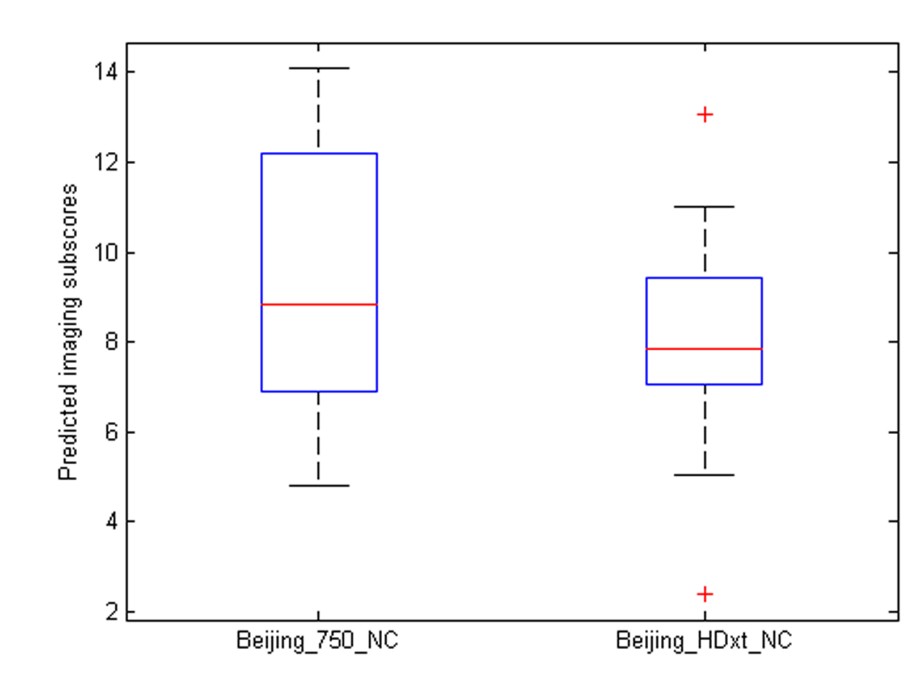

**Appendix 9—figure 1.** The distribution of the predicted imaging subscores of the healthy controls at different sites.

DOI: https://doi.org/10.7554/eLife.36173.040

Second, we investigated the relationships between the fMRI signal-to-noise ratio (SNR) and the predicted imaging subscores. Different MRI acquisition protocols (e.g. scanner hardware, imaging protocols and acquisition sequences) can influence the imaging SNR. But, it is not trivial to estimate the SNR in resting-state fMRI because the noise is complex and also differs spatially. Here, we calculated the temporal SNR (tSNR) as the ratio between the mean fMRI signal and its temporal standard deviation in each voxel (*Welvaert and Rosseel, 2013*), and then averaged across all voxels in each region of interest (ROI) (*Gardumi et al., 2016*; *Hay et al., 2017*). As there were 22 ROIs in this study, the median of these 22 ROI tSNR values was used as the measure for evaluating the SNR of the fMRI. We then correlated the median tSNR with the predicted imaging subscores across all of the NC subjects (n = 40), and found that there were significant correlations between them (Pearson's correlation r = 0.36, p=0.024) as shown in*Appendix 9—figure 2*.

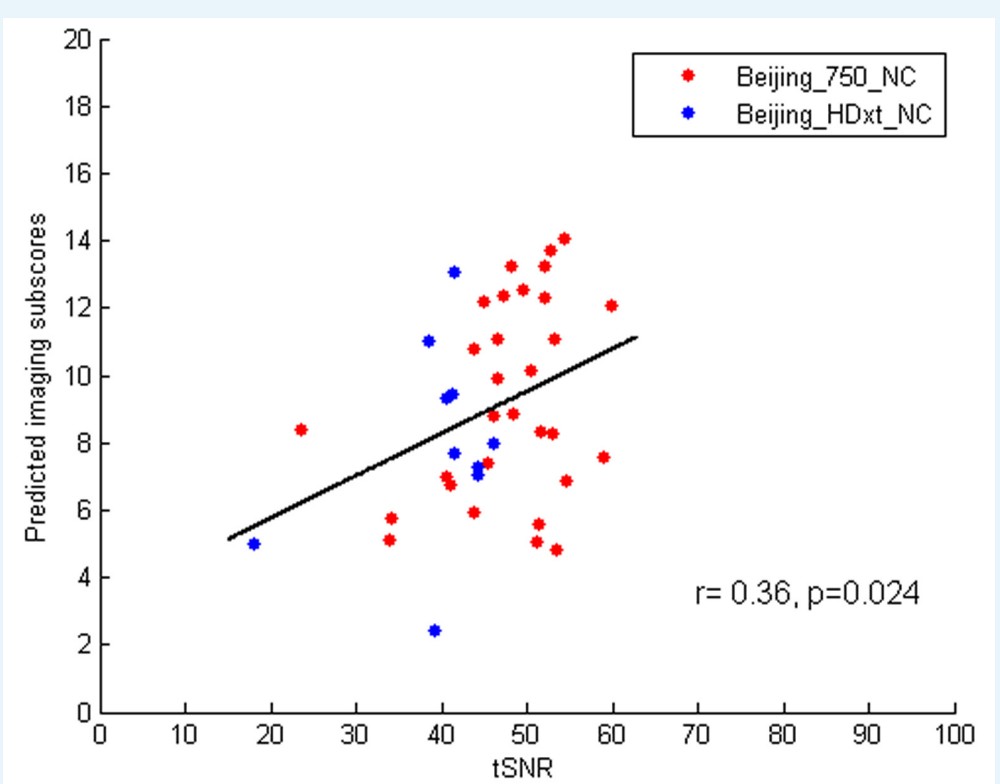

**Appendix 9—figure 2.** The correlations between the fMRI signal-to-noise ratio (SNR) and the predicted imaging subscores in the healthy controls.
DOI: https://doi.org/10.7554/eLife.36173.041

From the above two experiments, we found that (1) the fMRI tSNR could be one of influencing factors in the application of the presented model; (2) the predicted imaging subscores for the NC subjects could be approximate across different sites when the tSNR was proximity. Therefore, we suggested that our presented model can be applied to different centers, although the calibration might be required. Further, the tSNR in fMRI is not only associated with instrumental noise but also modulated by subject-related noise, such as physiological fluctuations and motion artifacts (*Huettel et al., 2009*). Therefore, we suggest that, on the one hand, the quality of imaging acquisition, including MRI scanner and imaging sequence/parameters, need to be guaranteed; and on the other hand, scanning protocols is required to be standardized to reduce the subject-related noise during the scanning.

