## [Decision Letter]

Thank you for submitting your article "Prognostication of chronic disorders of consciousness using brain functional networks and clinical characteristics" for consideration by *eLife*. Your article has been reviewed by Timothy Behrens as the Senior Editor, a Reviewing Editor, and three reviewers. The following individuals involved in review of your submission have agreed to reveal their identity: Klaas Enno Stephan (Reviewer #1); Adrian M Owen (Reviewer #2); Jessica Schrouff (Reviewer #3).

The reviewers have discussed the reviews with one another and the Reviewing Editor has drafted this decision to help you prepare a revised submission.

Summary:

This is a very interesting and timely study that examines the predictability of clinical outcome of patients with disorders of consciousness (DOC), based on "resting state" functional MRI and clinical data. It reports that both clinical data and neuroimaging data can separately predict individual clinical outcomes, but that the combined usage of both sources of information is superior.

The reviewers agreed that this study represents a potentially important attempt towards establishing practically useful predictors based on data that can be obtained in clinical practice. For DOC, the sample size is large, and the use of two separate test datasets is laudable. Having said this, we also identified several major concerns with regard to the statistical approach and the interpretation of the results. These are detailed below.

Essential revisions:

1) There are several concerns regarding the statistical approach taken. A main issue that featured in the discussion among reviewers is that feature selection and model performance assessment were performed without nested cross-validation. This should be done to obtain an accurate estimate of model performance in the training data set as well. In the present scenario in the paper, performance measures derived from the training set cannot be used as an estimate of the model's generalization ability. At present, the abstract and discussion are slightly misleading in that they report an average accuracy over training and validation sets. Second, the reasons for double feature selection (given the relatively low number of features) and, more generally, for the choice of PLSR are unclear. As the method is put forward in this work, comparison with simpler methods would be useful (e.g. vanilla SVM). Third, it is unclear how the bootstrap was used to perform internal validation. It only performs validation of the training samples. Typically, the trained model would be applied to the subjects not used in that bootstrap sample and the R2 would be computed based on those. Otherwise, the R2 value does not represent generalizability, but only goodness of fit. If this is a misunderstanding, this paragraph should be revised. Fourth, what regularisation was applied in PLSR? Fifth, there is little discussion of the PPV, while assessing the number of false negative is also very important. In all cases, statistical assessment through permutations of the labels is desired. Finally, when evaluating the performance of classification, it would be more convincing if you reported the balanced accuracy. This is because the results for the two different test datasets differ remarkably with respect to sensitivity (subsection “Accuracy comparison between single-domain models and combination model”); this suggests that the outcome groups were differentially balanced across the datasets.

2) The interpretation of classifier weights is problematic (Discussion section): Please see Haufe et al., 2014, Kia et al., 2017 and Weichwald et al., 2015 for the caveats of interpreting the weights of classification models. Especially, as feature selection is performed here, the authors could refrain from looking at the weights (and especially at their sign; for example, finding that duration of DOC has positive impact on outcome does not seem realistic) and instead identify which features were selected.

3) It is not clear that excluding patients who died is the right thing to do. At the very least, whether it was the right thing to do should have been discussed. On the one hand, I appreciate that if they died, one can never know if they would have regained consciousness. But in my experience (and I am sure this is the authors' experience too), these patients mostly die for a reason that is related to their initial injury (directly or indirectly). Plus, whether they die or not is of course part and parcel of the prognostication (Can we predict those who die?). I know the paper's intention was to try to predict recovery of consciousness (rather than who died), but I think an argument can be made for including those who died in the 'non recovery of consciousness' group. How does that affect the model? After all, a model that says, "Assuming you live, these data have an 88% chance of predicting whether you will regain consciousness" is nowhere near as good as a model that says, "These data have an 88% chance of predicting whether you will regain consciousness".

4) Subsection “MRI acquisition”. Supplementary material 1 shows quite a bit of variation in the time after injury at which the MRI was conducted (in the testing dataset alone, the range is 1 month to 77 months). What is noticeable is that almost all those who have a late scan do not recover. On the one hand, this is obvious – if you are still vegetative after 77 months you are very unlikely to recover whether you have a scan or not, and in this sense, the point of trying to 'predict recovery' is less obvious. Plus of course at this point there will have been many brain changes that do not reflect the original injury per se but may be the consequences of languishing in this stage for some years (atrophy, etc.). What would be most interesting would be to see how the model does with patients who, say are scanned within 3 months of their injury. These are the patients that we really care about because if we could prognosticate among them then it could really change clinical practice. To put it simply, including data from a patient who has been in a MCS for 77 months or VS for 54 months and predicting that they will not recover is not adding much (and may be affecting the model), because the chances of any meaningful recovery at that stage is so low anyway.

5) fMRI analysis: How were brain lesions taken into account? This could influence the functional connectivity measures considerably.

6) fMRI motion correction (subsection “Imaging preprocessing”): The application of band-pass filtering to "scrubbed" fMRI timeseries is not unproblematic given that the removal of scans from the time series can severely affect temporal autocorrelation. Also, were there significant differences in the number of scrubbed volumes between the outcome groups (for unresponsive patients, movement would seem to be less of an issue than for controls or more responsive patients), and could this have affected regression/classification? Did you consider using scan-nulling regressors as an alternative? (see, for example, the discussion in Caballero-Gaudes and Reynolds, 2017) In SM4, please split the graphs by outcome. If more movement was observed for one subtype of patients, this could drive all the observed results.

7) Subsection “Accuracy comparison between single-domain models and combination model”: When only the clinical features are included in the authors model the accuracy in the Guangzhou dataset is 83%. When both clinical and imaging features are included it is 88%. Given the rather small size of the sample, this seems to amount to a difference of one patient which is neither here nor there in terms of clinical prognostication. It's not wildly different for the other dataset either. Can you please comment on this? More generally, it seems overstated how much better the combined features did in the model – this would need a critical discussion but also better quantification. In other words, how much more is gained exactly by using both datasets as opposed to either dataset alone? Please report details of the statistical superiority of the combined feature set (p-value, effect size). Also, when you directly compare neuroimaging data and clinical data, is there any difference in how well they support prediction?

8) It is laudable that the code of the statistical model for predictions is made available. Will the code for the analysis pipeline and the data also be made available?

9) The structure and writing of the paper could be improved. In particular, some of the 'sign-posting' in the article is poor. For example, 'normal controls' are referred to in subsection “Subjects” and subsection “Definition of networks and regions of interest” and so on, but we never really learn much about them until the Results section. This had the unfortunate consequence of making the article feel very vague and lacking important details. They were mostly revealed much later (in the Supplementary material) but better sign posting earlier on would help with that. Likewise, in the Materials and methods section "The two testing datasets were then used to validate the model". This comes out of nowhere, and one has no idea what the authors were referring to at this point. Likewise, in the Materials and methods section "All patients were followed up at least 12 months after the T0 time point" sounds vague and under specified. A sign post to Supplementary material 1 would have really helped.

10) Discussion section. At least two of these three references (the ones that I am an author on) do not support the point being made at all. In neither Owen et al., 2006 nor Monti et al., 2010 was "recovery after many years" observed.

---

## [Author Response]

Essential revisions:1) There are several concerns regarding the statistical approach taken.A main issue that featured in the discussion among reviewers is that feature selection and model performance assessment were performed without nested cross-validation. This should be done to obtain an accurate estimate of model performance in the training data set as well. In the present scenario in the paper, performance measures derived from the training set cannot be used as an estimate of the model's generalization ability. At present, the abstract and discussion are slightly misleading in that they report an average accuracy over training and validation sets.

This study designed 257 features, i.e., 253 imaging features and 4 clinical features, while there were only 63 samples in the training dataset "Beijing 750". That is, the sample size was smaller than the number of features. Traditional k-fold cross validation could bring about a large prediction error. Therefore, we have provided an out-of-bag (OOB) estimation as an estimate of model performance in the training dataset (James et al., 2013). Specifically, for the original "Beijng_750" training dataset *x*, we left out one sample at a time and denoted the resulting sets by *x_(-1)_*,…, *x_(-n)_*. From each leave-one-out set *x_(-i)_*, we drew 1000 bootstrap learning sets of size *n-1*, built a prediction rule on every bootstrap learning set generated from *x_(-i)_*, and applied the rule to the test observation *x_i_*. A majority vote was then made to give a class prediction for observation *x_i_*. Double feature selection and class prediction based on the ROC curve were performed on each bootstrap sample, that is, only using the samples that did not have *xᵢ* in their bootstrap sample. OOB is able to give the mean prediction error on each training sample *xᵢ*. In this way, we were able to measure the prediction error of our method in the training dataset. Using the OOB estimation, we found that the prediction accuracy in the training dataset was up to 89%, and the sensitivity, specificity, positive predictive value (PPV) and negative predictive value (NPV) were 69%, 94%,100%, and 87%, respectively.

We have modified the relevant text in the Abstract and Discussion section. In addition, we have added some text in subsection “Internal validation of model” in the revised manuscript as follows.

"In addition, out-of-bag (OOB) estimation was used as an estimate of model classification performance in the training dataset (James et al., 2013). Specifically, for the original training dataset *x*, we left out one sample at a time and denoted the resulting sets by *x_(-1)_,…, x_(-n)_*. From each leave-one-out set *x_(-i)_*, 1000 bootstrap learning sets of size n-1 were drawn. On every bootstrap learning set generated from *x_(-i)_*, we carried out feature selection, built a PLSR regression and classification model, and applied the model on the test observation *x_i_*. A majority vote was then made to give a class prediction for observation *x_i_*. Finally, we calculated the accuracy for the whole training dataset *x*."

We have also added some text to subsection “Prognostic classification model and internal validation” in the revised manuscript as follows.

"The OOB was able to provide the mean prediction error on each training sample and estimate the generalizability of our method in the training dataset. Using the OOB estimation, we found that the prediction accuracy in the training dataset "Beijing 750" was 89%, and the sensitivity, specificity, PPV and NPV was 69%, 94%, 100%, and 87% respectively."

Second, the reasons for double feature selection (given the relatively low number of features) and, more generally, for the choice of PLSR are unclear. As the method is put forward in this work, comparison with simpler methods would be useful (e.g. vanilla SVM).

This study designed two types of imaging features from the resting state fMRI data. Thus, for each participant in this study, there were 231 (22×21/2) functional connectivity features and 22 brain area connection features. Although the total number of imaging features (i.e. 253) is low in comparison to the number of image voxels, it is more than the number of patients in the training dataset (i.e. 63 patients). In addition, we found that there was multi-collinearity in these 253 imaging features. The maximum correlation coefficient was up to 0.7985 (Pearson's correlation, p=4.5174e-015, uncorrected). Therefore, standard linear regression would fail in this study.

PLSR is a powerful and effective method to handle these sorts of problematic data sets. It can predict a response from a set of high-dimensional and collinear predictors. This prediction is achieved by extracting from the predictors a set of orthogonal factors called latent variables which have the best predictive power. PLSR has been widely used in a variety of fields (Wold et al., 2001), including fMRI data analysis (Krishnan et al., 2011). This is the reason that we chose to use PLSR in this study.

In addition, we hoped to use as few imaging features as possible in the predictive model, which could prove helpful to interpret and understand the model. We therefore carried out feature selection. Feature selection techniques have been widely adopted in brain analysis studies, in order to produce a small number of features for efficient classification or regression, and to reduce overfitting and increase the generalization performance of the model. Feature ranking and feature subset selection are two typical feature selection methods. Feature subset selection methods are generally time consuming, and even inapplicable when the number of features is extremely large, whereas ranking-based feature selection methods are subject to local optima. Therefore, these two feature selection methods are usually used jointly. Here, we first used a correlation-based feature ranking to select an initial set of features, and then adopted the CARS-PLSR method for further feature subset selection.

We have added some comparison with linear SVM. The code for SVM was downloaded from LIBSVM (https://www.csie.ntu.edu.tw/~cjlin/libsvm/). The 253 imaging features and the four clinical features were integrated into one feature vector. No feature selection was adopted in the linear SVM-based classification. The patients who recovered consciousness recovery were labeled as 1 (i.e. GOS score≥3), and those who did not were labeled as -1 (i.e. GOS score≤2).

(1) Performance in the training dataset

As outlined above in 1.1, the OOB estimation process was used to estimate the performance of the linear SVM in the training dataset "Beijing 750".

Training datasetNo feature selection +Linear SVMFeature selection + PLSRBeijing_750Accuracy: 83% Sensitivity: 31% Specificity: 96% PPV: 100% NPV: 81%Accuracy: 89% Sensitivity: 69% Specificity: 94% PPV: 100% NPV: 87%

(2) Performances in the two testing datasets

Using all of the samples in the training dataset "Bejing 750", we trained a linear SVM-based classifier and then tested the predictive accuracy in the two testing datasets.

Testing datasetNo feature selection +Linear SVMFeature selection + PLSRBeijing_HDxtAccuracy: 76% Sensitivity: 58% Specificity: 92% PPV: 88% NPV: 71%Accuracy: 88% Sensitivity: 83% Specificity: 92% PPV: 91% NPV: 86%Guangzhou_HDxtAccuracy: 88% Sensitivity: 100% Specificity: 83% PPV: 67% NPV: 100%Accuracy: 88% Sensitivity: 100% Specificity: 83% PPV: 67% NPV: 100%

Taking together the performance comparisons in both the training dataset and the two testing datasets, we believe that our method "Feature selection + PLSR" should be superior to " No feature selection + Linear SVM" in this study.

We have added the comparisons with linear SVM in subsection “Comparison between the proposed modeling method and linear SVM” as follows.

“We compared the prediction performances between the proposed modeling method and linear SVM. The code for SVM was downloaded from LIBSVM (https://www.csie.ntu.edu.tw/~cjlin/libsvm/). The 253 imaging features and the four clinical features were integrated into one feature vector. No feature selection was adopted in the linear SVM-based classification. The patients with GOS≥3 were labeled as 1, with the others being designated as -1 (i.e., GOS ≤2).

Similarly, the OOB estimation process was used to estimate the performance of linear SVM in the training dataset "Beijing 750". Next, using all the samples in the training dataset "Beijing 750", we trained a linear SVM-based classifier and then tested the predictive accuracy in the two testing datasets."

We have provided these comparison results in subsection “Comparison between the proposed modeling method and linear SVM” as follows.

"Using the OOB estimation, we found that the accuracy of the linear SVM-based classification method in the training dataset "Beijing 750" was 83% (sensitivity: 31%, specificity: 96%, PPV: 100%, NPV: 81%), which was lower than the accuracy of our proposed modeling method (i.e., accuracy: 89%, sensitivity: 69%, specificity: 94%, PPV: 100%, NPV: 87%). On the other hand, the linear SVM-based classification method achieved an accuracy of 76% (sensitivity: 58%, specificity: 92%, PPV: 88%, NPV: 71%) and 88% (sensitivity: 100%, specificity: 83%, PPV: 67%, NPV: 100%) in the "Beijing HDxt" testing dataset and the "Guangzhou HDxt" testing dataset, respectively. That is, the accuracy in the "Beijing HDxt" testing dataset was lower than that in our method, whereas the accuracy in the "Guangzhou HDxt" testing dataset was similar to that of our approach. Therefore, taking together the performance comparisons in both the training dataset and the two testing datasets, we believe that our method based on feature selection and PLSR should have higher prediction accuracy and better generalizability in comparison to linear SVM."

Third, it is unclear how the bootstrap was used to perform internal validation. It only performs validation of the training samples. Typically, the trained model would be applied to the subjects not used in that bootstrap sample and the R2 would be computed based on those. Otherwise, the R2 value does not represent generalizability, but only goodness of fit. If this is a misunderstanding, this paragraph should be revised.

The internal validation in this study was based on the proposal of the Prognosis Research Strategy (PROGRESS) group (Steyerberg, et al., 2013). The procedure was strictly implemented according to Steyerberg, (2008).

The main goal of internal validation is to correct for overfitting, which causes optimism about a model's performance in new subjects. In the chapter 5 of Steyerberg, (2008), optimism is defined as true performance minus apparent performance, where true performance refers to the performance in the underlying population, and apparent performance refers to the estimated performance in the sample. Bootstrapping can be used to quantify the optimism of a prediction model. This optimism is subsequently subtracted from the original estimate to obtain an “optimism-corrected” performance estimate. The formula is as follows.

Optimism-corrected performance is calculated as:

Optimism-corrected performance = Apparent performance in sample − Optimism,

where

Optimism = Bootstrap performance − Test performance.

The exact steps are as follows:

1) Construct a PLSR model in the training dataset "Beijing_750"; determine the apparent coefficient of determination *R^2^* on the training dataset;

2) Draw a bootstrap sample (denoted by *Sample_*_*) with replacement from the original training sample (denoted by *Sample*);

3) Construct a model (*Model_*_*) in *Sample_*_*, replaying every step that was done in the original *Sample*, including the double feature selection and PLSR. Determine the bootstrap coefficient of determination *R*_*_*^2^* as the apparent performance of *Model_*_* on *Sample_*_*;

4) Apply *Model_*_* to the original *Sample* without any modification to determine the test performance *R_T_^2^*;

5) Calculate the optimism as the difference between bootstrap performance *R*_*_*^2^*and test performance *R_T_^2^*, that is, *Optimism = R*_*_*^2^ − R_T_^2^*;

6) Repeat steps 1–4 1000 times, to obtain a stable estimate of the optimism;

7) Subtract the optimism estimate (step 5) from the apparent performance *R^2^* (step 1) to obtain the optimism-corrected performance estimate.

To make the description more precise and clearer, we have modified the text in in subsection “Internal validation of model” as follows.

"The prognostic regression model was internally validated using bootstrap (Steyerberg, 2008). Specifically, bootstrap samples were drawn with replacement from the training dataset "Beijing 750" such that each bootstrap sampling set had a number of observations equal to that of the training dataset. Using a bootstrap sampling set, the correlation-based feature selection and CARS-PLSR was first used to select feature subset, after which the PLSR was used to generate a prognostic model. We then applied the model to the bootstrap sampling set and the original training dataset and calculated the coefficient of determination *R^2^* of each of the two datasets. The difference between the two coefficients of determination was defined as the optimism. The process was repeated 1000 times to obtain a stable estimate of the optimism. Finally, we subtracted the optimism estimate from the coefficient of determination *R^2^* on the "Beijing 750" training dataset to obtain the optimism-corrected performance estimate."

Fourth, what regularisation was applied in PLSR?

We did not use any regularisation in PLSR. There is only one hyperparameter in our method, that is, the number of latent variables. Given that nine predictors were involved in the PLSR, the rank of the predictors matrix *X* was less than or equal to nine, which meant that the number of latent variables could range from 1 to 8. We used cross-validation to decide that the number of latent variables was 3. We also found that the classification results were not sensitive to different number of latent variables in our study.

Fifth, there is little discussion of the PPV, while assessing the number of false negative is also very important. In all cases, statistical assessment through permutations of the labels is desired.

The PPV across the two testing datasets was up to 91% and 67%, respectively. That is, the PPV for the Guangzhou dataset was relatively low. Our method predicted that nine patients in this dataset would recover consciousness recovery, but only six of them did (i.e. GOS score≥3), with the other three patients remaining unconsciousness at the end of the follow-up. In the two testing datasets, our method successfully identified the 16 out of the total 18 patients (16/18) who later recovered consciousness. That is, the number of false negatives was two.

We have added some discussion in Discussion section as follows.

"Secondly, the PPVs for the two testing datasets was remarkably different, with that for the "Guangzhou HDxt" dataset being relatively low (67% versus 91%). Although our method predicted that nine patients in this dataset would recover consciousness, only six of them did (i.e. GOS≥3), with the other three remaining unconscious at the end of the follow-up (i.e. GOS≤2). Given that many additional factors are associated with the outcome of DOC patients, including medical complications, nutrition and so on, future work will need to integrate more information in order to provide more precise predictions."

We have also provided the statistical assessment based on the permutation test for the classification accuracies in the two testing datasets. Specifically, we repeated the classification procedure after randomizing the labels of the testing dataset 1000 times. The p-value was then given by the percentage of runs for which the classification accuracy obtained was greater than the classification accuracy obtained in the dataset with the original label. The results were as follows.

1) The "Beijing_HDxt" testing dataset

In the "Beijing HDxt" dataset, the prediction accuracy in the dataset with the original label was 88%. We found that it was significantly greater than the classification accuracy after randomizing the labels (permutation test, p<0.001).

2) The "Guangzhou_HDxt" testing dataset

In the "Guangzhou HDxt" dataset, the prediction accuracy in the dataset with original label was 88%. We found that it was also significantly greater than the classification accuracy after randomizing the labels (permutation test, p<0.001).

**Author response image 2. respfig2:** 

Finally, when evaluating the performance of classification, it would be more convincing if you reported the balanced accuracy. This is because the results for the two different test datasets differ remarkably with respect to sensitivity (subsection “Accuracy comparison between single-domain models and combination model”); this suggests that the outcome groups were differentially balanced across the datasets.

We have acknowledged that the outcomes for the two testing datasets were remarkably different. In the "Beijing_HDxt" dataset, 12 out of the total 25 patients recovered consciousness, while six out of the total 24 patients recovered consciousness in the "Guangzhou_HDxt" dataset. In fact, the etiology for the patients was remarkably different in the two datasets. The number of patients in trauma/stroke/anoxia cohorts were 12/6/7 and 8/0/16 in the "Beijing_HDxt" and "Guangzhou_HDxt" dataset, respectively. Given that the characteristics of the two medical centers and their roles in the local health care system are different, we speculate that the enrolled patient populations might be different, which could be one reason why the outcome groups were differentially balanced across the two datasets. However, we would like to emphasize that these etiologies are the main sources of DOC, and a good prognostication method should be applicable to different etiologies and to different datasets from different medical centers.

Using the predicted scores of the patients in the two testing datasets, we plotted the ROC curves for each of the two testing datasets as follows.

**Author response image 3. respfig3:** 

From the above figure, we can see that the AUC for each of the two testing datasets is larger than 0.9, although there are some differences between them in terms of the ROC.

2) The interpretation of classifier weights is problematic (Discussion section): Please see Haufe et al., 2014, Kia et al., 2017 and Weichwald et al., 2015 for the caveats of interpreting the weights of classification models. Especially, as feature selection is performed here, the authors could refrain from looking at the weights (and especially at their sign; for example, finding that duration of DOC has positive impact on outcome does not seem realistic) and instead identify which features were selected.

We believe that the interpretation of "classifier weights" that the reviewer mentioned should refer to the regression weights in the PLSR.

According to the reviewer's suggestion, we reviewed the studies on the interpretation of weights of linear models in neuroimaging analysis. These studies, including Haufe et al., (2014), Kia et al., (2017) and Weichwald et al., (2015), are very instructive. They illustrated the source of the problem and provided some general discussions and heuristic rules. In addition, we reviewed some specific literatures focusing on the interpretation of predictor importance in PLSR. Given that model interpretation is an important task in most applications of PLSR, researchers have been looking for optimal interpretation methods and have made considerable progress (Kvalheim and Karstang, 1989; Kvalheim et al., 2014; Tran et al., 2014).

In summary, as the reviewer pointed out, directly interpreting the model using model parameters such as the weights and loadings in the PLSR is a problematic exercise. Therefore, we have used one recent method (Tran et al., 2014), i.e. significance Multivariate Correlation (sMC), to analyze the predictor importance in our PLSR model. We have corrected the relevant results and added some discussions to the manuscript. In the following, we will firstly elaborate the cause of problem, and then introduce the sMC method for interpreting variable importance in PLSR. After that, we will present our new results.

Cause of problem:

In Haufe et al., (2014), it was demonstrated that linear backward models, i.e. linear decoding models, cannot be interpreted in terms of weight, as a large weight does not necessarily correspond to a feature that picks up the signal and, vice-versa, a feature that picks up the signal does not necessarily have a large weight. For example, consider a simple regression case with two predictors, *X* = [*x_1_(n) χ2(n)*], and a response *y(n)*. Imagine that *y(n)=signal(n)*, and *x_1_(n)=signal(n)-2*noise(n)* and *χ2(n)=noise(n)*, where the *noise(n)* is random noise and independent of the *signal(n)*. Thus, we can easily get *y(n) = x_1_(n)* +2* *χ2(n)*. The weight for *χ2(n)* is larger than the weight for *x_1_(n)*. However, it is evident that *χ2(n)* is just noise and could be any value. In this case, the weight does not reflect variable importance. In fact, the weights are determined not only by the correlation of *X* to the response *y*, but also concurrently by the covariance in *X*.

Therefore, some researchers suggest that the lack of interpretability of multivariate decoding in neuroimaging could be a direct consequence of low signal-to-noise ratios, the high dimensionality of neuroimaging, high correlations among different dimensions of data, and cross-subject variability (Kia et al., 2017).

Introduction of the sMC method:

The goal of PLSR is to predict *Y* from *X* and to describe their common structure (Krishnan et al., 2011). It finds latent variables stored in a matrix *T* that model *X* and simultaneously predict *Y*. Formally this is expressed as a double decomposition of *X* and the predicted Y^:

X=TPTandY^=TBCT (Eq. 1)

where *P* and *C* are loadings (or weights) and *B* is a diagonal matrix. These latent variables are ordered according to the amount of variance of Y^ that they explain. Rewriting Eq. (1) shows that Y^ can also be expressed as a regression model as:

Y^=TBCT=XBPLS (Eq. 2)

with

BPLS=PT+BCT (eq. 3)

where PT+ is the Moore-Penrose pseudo-inverse of PT. The matrix BPLS is equivalent to the regression weights of multiple regression. From the above equations, we can see that PLSR finds components from *X* that are also relevant for *Y*. Specifically, PLSR searches for a set of components (called latent vectors) that performs a simultaneous decomposition of *X* and *Y* with the constraint that these components explain as much as possible of the covariance between *X* and *Y*. Therefore, PLSR is robust to the noise in comparison to standard linear regression.

However, PLSR constructs latent variables that not only maximize the correlation of *X* to the response *y*, but also concurrently try to maximize the explained variance in *X*. When the dominant source of variation is not related to *y*, the maximization of the explained *X*-variance is likely to bring irrelevant information into the PLSR model, meaning that the interpretation of PLSR model parameters and variable importance based on these parameters is not a straightforward exercise. The literature largely discusses this issue and multiple approaches have been proposed to tackle the problem (Kvalheim and Karstang, 1989; Kvalheim et al., 2014; Tran et al., 2014).

Using the property of the basic sequence theory, Tran et al., (2014) developed a method called sMC to statistically assess variable importance in PLSR. The key points in sMC are to estimate for each variable the correct sources of variability resulting from the PLSR (i.e. regression variance and residual variance) and use them to statistically determine a variable's importance with respect to the regression model. In sMC, the F-test values (called sMC F-values) are used to evaluate the predictors' importance in the prognostic model. In the paper by Tran et al., (2014), the performance of sMC was demonstrated on both an omics-related simulated dataset and a real dataset of NIR spectra of pharmaceutical tablets.

A simulation to test the sMC method:

Let us return to the above regression example. For a clearer explanation, we give certain values to the X, y, and *noise*, and then perform some simulations in Matlab. The code is shown in Author response image 4 below.

**Author response image 4. respfig4:** 

We specified different amplitude, i.e. *noise_amplitude*, to obtain the results at different noise levels:

**Author response image 5. respfig5:** 

**Author response image 6. respfig6:** 

**Author response image 7. respfig7:** 

**Author response image 8. respfig8:** 

Although the above simulation is a simple regression example, we can see that the sMC F-value can robustly discriminate the importance of the two predictors even when the amplitude of random noise is 50 times the standard deviation of the signal.

Our results:

We used the sMC method to analyze the predictor importance in our study. The results are as follows.

**Author response image 9. respfig9:** 

In the above figure, the x-axis represents each predictor, and the y-axis is the sMC F-value. The first~fifth predictors are the imaging predictors, that is, DMN.aMPFC, DMN.PCC, ExecuContr.DMPFC, DMN.aMPFC – ExecuContr.DMPFC, Auditory.MCC – Visual.R.V1, respectively. The sixth-ninth predictors are the clinical predictors, that is, Incidence_age, Duration_of_DOC, Etiology_traumatic, Etiology_anoxic, respectively.

We have corrected Figure 4 and modified the relevant text in the manuscript. We have also deleted the discussion about the sign of regression weights.

Finally, we would like to refer to the paper of Kia et al., (2017), who stated that " Intuitively, the interpretability of a brain decoder refers to the level of information that can be reliably derived by an expert from the resulting maps…". Our prognostication model can be reliably applied to independent testing datasets, and the predictors can be basically explained by previous studies and experimental data. Therefore, we suggest that the prognostication model should be interpretable.

3) It is not clear that excluding patients who died is the right thing to do. At the very least, whether it was the right thing to do should have been discussed. On the one hand, I appreciate that if they died, one can never know if they would have regained consciousness. But in my experience (and I am sure this is the authors' experience too), these patients mostly die for a reason that is related to their initial injury (directly or indirectly). Plus, whether they die or not is of course part and parcel of the prognostication (Can we predict those who die?). I know the paper's intention was to try to predict recovery of consciousness (rather than who died), but I think an argument can be made for including those who died in the 'non recovery of consciousness' group. How does that affect the model? After all, a model that says, "Assuming you live, these data have an 88% chance of predicting whether you will regain consciousness" is nowhere near as good as a model that says, "These data have an 88% chance of predicting whether you will regain consciousness".

The reviewer raises an important point. However, if the patients died, we could not assess their CRS-R scores at the end of the follow-up. In this study, the development of the regression model was based on this. In the current situation, as the number of training samples was relatively low, it is difficult to involve those patients who died as the "missing data" in the regression model. On the other hand, we used the present model to test the patients who died. However, we are uncertain whether or not this work makes sense. After all, the present model is developed based on the presence or absence of consciousness recovery. We have acknowledged that mortality is one outcome of DOC, but this study has not involved this aspect. We have added some discussion in the Discussion section of the revised manuscript to address this:

"Finally, a DOC patient’s prognosis can be considered according to at least three dimensions: survival/mortality, recovery of consciousness, and functional recovery. This study focused on predicting recovery of consciousness, and the patients who died during the follow-up were excluded. In the future, we will consider more outcome assessments, including survival/mortality and functional recovery."

4) Subsection “MRI acquisition”. Supplementary material 1 shows quite a bit of variation in the time after injury at which the MRI was conducted (in the testing dataset alone, the range is 1 month to 77 months). What is noticeable is that almost all those who have a late scan do not recover. On the one hand, this is obvious – if you are still vegetative after 77 months you are very unlikely to recover whether you have a scan or not, and in this sense, the point of trying to 'predict recovery' is less obvious. Plus of course at this point there will have been many brain changes that do not reflect the original injury per se but may be the consequences of languishing in this stage for some years (atrophy, etc.). What would be most interesting would be to see how the model does with patients who, say are scanned within 3 months of their injury. These are the patients that we really care about because if we could prognosticate among them then it could really change clinical practice. To put it simply, including data from a patient who has been in a MCS for 77 months or VS for 54 months and predicting that they will not recover is not adding much (and may be affecting the model), because the chances of any meaningful recovery at that stage is so low anyway.

Although the range of the times after injury at which the MRI scanning occurred (i.e., the T0 timepoint) was relatively wide (1-77 months), the majority (i.e., 65 out of 112) of the patients were scanned no more than 3 months after the injury, so the median of T0 was equal to or less than 3 months as shown in Table 1 in the manuscript. We emphasize that this is clinically-useful time point and close to the time of injury.

Notably, our model demonstrated good sensitivity and specificity in both the "subacute" patients (i.e. duration of unconsciousness ≤3 months) and in those in the chronic phase (i.e. duration of unconsciousness >3 months), as shown in Figure 7 of the manuscript.

**Author response image 10. respfig10:** 

We also found that our model correctly predicted the nine out of ten patients with a DOC duration longer than or equal to three months who subsequently recovered consciousness, including three patients with longer than or equal to six months duration. This suggested that our model might be feasible to assess prolonged DOC patients.

As DOC can be acute or chronic, the diagnosis and prognosis for prolonged patients is valuable for both the patients and their families and caregivers (Bernat, 2006). The social, economic, and ethical consequences associated with DOC are tremendous. Neurologists are often confronted with the expectations of the prolonged patients' families. Accordingly, several highly impactful functional imaging studies have enrolled the prolonged DOC patients, for example, the patients with 24 years disease duration published in Lancet (Stender et al., 2014), and the ones with 5 years disease duration (Vanhaudenhuyse et al., 2010) or over 25 years disease duration (Demertzi et al., 2015) published in Brain. A recent study published in Brain also enrolled prolonged DOC patients (Chennu et al., 2017).

A prolonged state of impaired consciousness is devastating. It has been widely considered that a clinically meaningful recovery is unrealistic for prolonged DOC patients (Wijdicks and Cranford, 2005). However, recent pilot studies challenge the existing practice of early treatment discontinuation for these patients, and motivate further research to develop therapeutic interventions, although more validations are required. For example, one study published in Nature reported the use of deep brain stimulation in a MCS patient with 6 years disease duration (Schiff et al., 2007), and another study published in Current Biology reported the use of vagus nerve stimulation in a VS/UWS patient with 15 years disease duration (Corazzol et al., 2017). Recently, we reported the first case of transcutaneous auricular vagus nerve stimulation in DOC (Yu et al., 2017).

However, before using a novel therapeutic intervention, clinicians first need to determine if the patient is a suitable candidate for these treatments. This requires thorough examination and a review of possible outcomes prior to any intervention. Therefore, an accurate and robust prognostic test for the DOC patient, including a prolonged one, is of significant clinical value. We believe that our study has made important progress in this direction.

5) fMRI analysis: How were brain lesions taken into account? This could influence the functional connectivity measures considerably.

According to the clinical characteristics of DOC, this study has carefully taken into account the lesions of the DOC patients during the imaging preprocessing.

The etiology of DOC can be subdivided into three types, including trauma, stroke and anoxic brain injury. The three causes often have different locations of injury and distinct severities of damage to brain structure and function. For example, for trauma, brain injury is mainly from shear, oppression, and impact, so the DOC patients with trauma could have large or small brain deformation or an incomplete brain. The injury can be relatively limited in a focal region or diffused axonal damage. In stroke, the patient's brain is relatively complete, and the injury location is generally limited to the blood supply area. In anoxia, the brain is always complete, but the injury diffuses through the whole cortex.

In this study, we checked the T1-weighted 3D high-resolution image for each patient. According to the size of the visible lesion in the T1 image, we divided the lesion into large focal lesions, small focal lesions, and diffused lesions. Large focal lesions often destroy the spatial warping of the ROIs, violate the registration performance and further disrupt the data analysis. Previous studies (Di Perri et al., 2016) used the exclusion criterion that large focal brain damage occupied more than two-thirds of one hemisphere. Our study referred to this criterion and excluded the patient with large focal lesions that exceeded 30% of the total brain volume. In contrast, small focal lesions and diffused lesions should have very limited effects on the registration. ROI-based functional connectivity analysis has been widely used in brain disease studies (Zhang and Raichle, 2010), including DOC studies (Demertzi et al., 2015; Qin et al., 2015; Silva et al., 2015). These brain lesions can not only influence the functional connectivity of the local ROI in which the lesion was located in but also cause the widespread changes throughout the brain (Alstott et al., 2009). Here, we hypothesize that such abnormal manifestations in functional integration from small focal lesions and diffused lesions could be promising markers for the prognosis of DOC patients.

This study developed protocols to segment the lesion and calculate the volume. Specifically, three authors (MS, YY, and JHH) completed segmentation of the focal lesion and volume calculation. First, using ITK-SNAP and in-house software, the author (MS) segmented the focal lesion in the T1-weighted 3D high-resolution image. YY and JHH, who are both experienced neurosurgeon, then visually checked the segmented lesion slice by slice and modified the boundary if necessary. In this way, focal lesions were extracted from the high-resolution T1 images, and the volume of the lesions was computed. The following figure shows an example ("Beijing_750" dataset, patient alias='013', diagnosis=VS/UWS, GOS=3). The lesion volume within the left brain hemisphere was about 73000ml, 7% of the total brain volume (i.e., the summary of gray matter and white matter).

**Author response image 11. respfig11:** 

We also developed a toolkit to visually check the transformed ROIs for each patient. More examples are provided in Appendix 5 and Supplementary file 1.

6) fMRI motion correction (subsection “Imaging preprocessing”): The application of band-pass filtering to "scrubbed" fMRI timeseries is not unproblematic given that the removal of scans from the time series can severely affect temporal autocorrelation. Also, were there significant differences in the number of scrubbed volumes between the outcome groups (for unresponsive patients, movement would seem to be less of an issue than for controls or more responsive patients), and could this have affected regression/classification? Did you consider using scan-nulling regressors as an alternative? (see, for example, the discussion in Caballero-Gaudes and Reynolds, 2017) In SM4, please split the graphs by outcome. If more movement was observed for one subtype of patients, this could drive all the observed results.

Motion artifact may produce a wide variety of signal changes in the fMRI data, and thus introduce complicated shifts in functional connectivity analysis. This problem is particularly serious for patients with brain diseases, as they are unlikely to follow the experimental instructions well and control their head motion. It is common practice to exclude patients who exhibit excessive head motion, but this would result in only a few patients being enrolled in studies such as ours, and the statistical power could therefore be significantly reduced. Power et al., (2012, 2015) introduced a novel method, called “scrubbing”, that identifies motion-induced spikes in the rsfc-MRI timeseries and excises these data with a temporal mask; adjacent timepoints are then temporally concatenated. Subsequently, Carp (2013) proposed a modification of scrubbing where data were removed and interpolated prior to band-pass filtering in order to avoid propagation of the motion artifact during the application of the band-pass filter. Using simulated data, he demonstrated that this modified scrubbing procedure was able to recover the “ground truth” connectivity in the timeseries (Carp, 2013). This kind of "scrubbing" method has proven one of the most popular approaches for taking the head movement into account, and a great many clinical rs-fMRI studies have adopted the approach, for example, in depression (Drysdale et al., 2017), Parkinson's disease (Mathys et al., 2016), stroke (Adhikari et al., 2017), traumatic brain injury (Dall'Acqua et al., 2017), and the DOC (Qin et al., 2015; Wu et al., 2015). In the paper by Caballero-Gaudes and Reynolds (2017), the authors acknowledged that "scrubbing" is "a popular approach" and proposed it to be "equivalent to adding scan nulling regressors in the model". Therefore, we used the method here to reduce the motion artifacts of the DOC patients.

We did not found any significant differences in the number of remaining volumes after scrubbing between the "consciousness recovery" group (GOS≥3) and "consciousness nonrecovery" group (GOS≤2) in either the training dataset or in the two testing datasets (two-sample Kolmogorov-Smirnov test, p=0.21, 0.51, 0.14 in the training dataset "Beijing 750", the testing dataset "Beijing_HDxt" and "Guangzhou HDxt", respectively). In addition, we found that there were no significant correlations between the number of remaining volumes and the predicted scores in the training dataset and the two testing datasets (Pearson's correlation, p>0.05). The following figures shows the relationship between the number of remaining volumes and the predicted scores and the predicted imaging subscores in the training dataset. Therefore, we do not think that the number of volumes remaining after scrubbing would significantly affect the regression and classification in our study.

**Author response image 12. respfig12:** 

**Author response image 13. respfig13:** 

However, in response to the reviewer's suggestion, we have split the graphs in Appendix 4 based on outcome. Please see the new Appendix 4.

In summary, as the reviewer pointed out, scrubbing can destroy the temporal structure of the fMRI data, but to balance the demands of noise reduction and data preservation and to follow the well-accepted preprocessing method in literature, our study has to adopt this compromise "scrubbing" approach to reduce motion artifacts.

7) Subsection “Accuracy comparison between single-domain models and combination model”: When only the clinical features are included in the authors model the accuracy in the Guangzhou dataset is 83%. When both clinical and imaging features are included it is 88%. Given the rather small size of the sample, this seems to amount to a difference of one patient which is neither here nor there in terms of clinical prognostication. It's not wildly different for the other dataset either. Can you please comment on this? More generally, it seems overstated how much better the combined features did in the model – this would need a critical discussion but also better quantification. In other words, how much more is gained exactly by using both datasets as opposed to either dataset alone? Please report details of the statistical superiority of the combined feature set (p-value, effect size). Also, when you directly compare neuroimaging data and clinical data, is there any difference in how well they support prediction?

To test whether the combined use of clinical and neuroimaging data is superior to that of any single domain data, we have added some new results to the revised manuscript. We resampled with replacement from the training dataset and built a regression and classification model based on clinical features alone, neuroimaging features alone, or a combination of the two-domain features. We then tested the classification accuracy in the two testing datasets using the three predictive models. In this way, we obtained the distribution of the prediction accuracies using each of the three types of the models. Next, we used repeated measures ANOVA to test whether or not the performances of the three types of models were the same, as well as Ψ, the root-mean-square standardized effect, to report the effect sizes of the mean differences between them. The effect size analysis was carried out using the Matlab toolbox "Measures of Effect Size" (https://github.com/hhentschke/measures-of-effect-size-toolbox).

1) The "Beijing_HDxt" testing dataset

**Author response image 14. respfig14:** 

Using repeated measures ANOVA, we found that there were significant differences between the mean of the classification accuracies using the three types of models (p<0.001). We subsequently conducted pairwise comparisons and found that there was a significant difference between the combination model and the model that used imaging features alone (paired sample t-test, p=0.001) or clinical features alone (paired sample t-test, p<0.001). We also found that there was significant difference between the model using imaging features alone and the model using clinical features alone (paired sample t-test, p<0.001).

Using effect size analysis, we found that there was a mean difference of Ψ=0.004 (95% confidence intervals [0.002, 0.007]) between the combination model and the model using imaging features alone, and Ψ=0.149 (95% confidence intervals [0.147, 0.152]) between the combination method and the model using clinical features alone. We also found that there was a mean difference of Ψ=0.145 (95% confidence intervals [0.142, 0.147]) between the methods using imaging features alone and clinical features alone.

2) The "Guangzhou_HDxt" testing dataset

**Author response image 15. respfig15:** 

Using repeated measures ANOVA, we found that there were significant differences between the mean of the classification accuracies using the three types of models (p<0.001). We subsequently conduct pairwise comparisons and found that there was a significant difference between the combination model and the model using imaging features alone (paired sample t-test, p<0.001) or clinical features alone (paired sample t-test, p<0.001). We also found that there was a significant difference between the model using imaging features alone and that only using clinical features (paired sample t-test, p<0.001).

Using effect size analysis, we found that there was a mean difference of Ψ=0.080 (95% confidence intervals [0.076, 0.084]) between the combination model and the model using imaging features alone, and a mean difference of Ψ=0.034 (95% confidence intervals [0.028, 0.040]) between the combination model and the model using clinical features alone. We also found that there was a mean difference of Ψ= -0.046 (95% confidence intervals [-0.053, -0.040]) between the model using only imaging features and that using only clinical features.

3) Summary

Firstly, in both the two testing datasets, the combination of imaging and clinical features demonstrated better performances than using imaging features alone or using clinical features alone. Secondly, the mean differences between the models using neuroimaging features alone or using clinical features alone were reversed in the two testing datasets, which suggested that the two testing datasets were heterogeneous and that it is necessary to integrate different information to improve prognostication for DOC.

We have provided these statistical results in the subsection "Comparison between single-domain model and combination model”.

In addition, we have added some discussion in the Discussion section as follows.

"This study involved two testing datasets, which were found to be quite different, as shown in Table 1. First, the distributions of the etiology of the patients were remarkably different in the two datasets. The numbers of patients suffering from trauma/stroke/anoxia were 12/6/7 and 8/0/16 in the "Beijing HDxt" and "Guangzhou HDxt" datasets, respectively. The outcomes were also different. In the "Beijing HDxt" dataset, 12 out of the total 25 patients recovered consciousness, compared with six out of the total 24 patients in the "Guangzhou HDxt" dataset. Given that the characteristics of the two medical centers and their roles in the local health care system are different, we speculate that this could be one of the main reasons that the enrolled patient populations were heterogeneous. As described in the Introduction, DOC can have many causes and be associated with several neuropathological processes and different severities, leading to the suggestion that information from different domains should be integrated to improve diagnosis and prognostication (Bernat, 2016). Our study demonstrates that the combination of imaging and clinical features can achieve a better performance than the use of single domain features."

8) It is laudable that the code of the statistical model for predictions is made available. Will the code for the analysis pipeline and the data also be made available?

We have opened the analysis pipeline in the package "pDOC", including fMRI preprocessing, feature calculation and extraction, regression and classification, and results visualization. Anyone is welcome to download it from the github (https://github.com/realmsong504/pDOC). Appendix 1 included anonymised demographic and clinical characteristics for each enrolled patient. However, there is no current plan to release the patients' imaging data due to confidentiality agreements.

9) The structure and writing of the paper could be improved. In particular, some of the 'sign-posting' in the article is poor. For example, 'normal controls' are referred to in subsection “Subjects” and subsection “Definition of networks and regions of interest” and so on, but we never really learn much about them until the Results section. This had the unfortunate consequence of making the article feel very vague and lacking important details. They were mostly revealed much later (in the Supplementary material) but better sign posting earlier on would help with that. Likewise, in the Msterials and methods section "The two testing datasets were then used to validate the model". This comes out of nowhere, and one has no idea what the authors were referring to at this point. Likewise, in the Materials and methods section "All patients were followed up at least 12 months after the T0 time point" sounds vague and under specified. A sign post to Supplementary material 1 would have really helped.

We have carefully revised the manuscript in light of these comments. (please see Materials and methods section). In addition, we have carefully proofread the manuscript.

10) Discussion section. At least two of these three references (the ones that I am an author on) do not support the point being made at all. In neither Owen et al., 2006 nor Monti et al., 2010 was "recovery after many years" observed.

We thank the reviewer for pointing this out. We have modified the text in the Discussion section as follows.

"Some influential studies have found that a few VS/UWS patients exhibit near-normal high-level cortical activation in response to certain stimuli or command (Owen et al., 2006; Monti et al., 2010), and that late recovery of responsiveness and consciousness is not exceptional in patients with VS/UWS (Estraneo et al., 2010). "

Additional references:

Adhikari MH, Hacker CD, Siegel JS, Griffa A, Hagmann P, Deco G, et al.

Decreased integration and information capacity in stroke measured by whole

brain models of resting state activity. Brain 2017; 140: 1068-85.

Caballero-Gaudes C, Reynolds RC. Methods for cleaning the BOLD fMRI signal.

Neuroimage 2017; 154: 128-49.

Carp J. Optimizing the order of operations for movement scrubbing: Comment on Power et al. Neuroimage 2013; 76(1): 436-8.

Dall'Acqua P, Johannes S, Mica L, Simmen HP, Glaab R, Fandino J, et al.

Functional and structural network recovery after mild traumatic brain injury: A

1-year longitudinal study. Frontiers in Human Neuroscience 2017; 11.

Drysdale AT, Grosenick L, Downar J, Dunlop K, Mansouri F, Meng Y, et al.

Resting-state connectivity biomarkers define neurophysiological subtypes of

depression. Nature Medicine 2017; 23(1):28-38

Haufe S, Meinecke F, Goergen K, Daehne S, Haynes J-D, Blankertz B, et al. On the interpretation of weight vectors of linear models in multivariate neuroimaging. Neuroimage 2014; 87: 96-110. 18 / 37

Kia SM, Pons SV, Weisz N, Passerini A. Interpretability of multivariate brain maps in linear brain decoding: definition, and heuristic quantification in multivariate analysis of MEG time-locked effects. Frontiers in Neuroscience 2017; 10: 22.

Mathys C, Caspers J, Langner R, Sudmeyer M, Grefkes C, Reetz K, et al.

Functional connectivity differences of the subthalamic Nucleus related to

Parkinson's disease. Human Brain Mapping 2016; 37(3): 1235-53.

Power JD, Barnes KA, Snyder AZ, Schlaggar BL, Petersen SE. Spurious but

systematic correlations in functional connectivity MRI networks arise from

subject motion. Neuroimage 2012; 59(3): 2142-54.

Weichwald S, Meyer T, Ozdenizci O, Schoelkopf B, Ball T, Grosse-Wentrup M. Causal interpretation rules for encoding and decoding models in neuroimaging. Neuroimage 2015; 110: 48-59.